# From Layers to States: A State Space Model Perspective to Deep Neural Network Layer Dynamics

**Qinshuo Liu**[*1], **Weiqin Zhao**[*1], **Wei Huang**[2], **Yanwen Fang**[1], **Lequan Yu**[†1], **Guodong Li**[†1]

[1]School of Computing and Data Science, The University of Hong Kong
[2]Department of Electrical and Electronic Engineering, The University of Hong Kong
{u3008680, wqzhao98, u3545683}@connect.hku.hk
aaron.weihuang@gmail.com
{lqyu, gdli}@hku.hk

## Abstract

The depth of neural networks is a critical factor for their capability, with deeper models often demonstrating superior performance. Motivated by this, significant efforts have been made to enhance layer aggregation - reusing information from previous layers to better extract features at the current layer, to improve the representational power of deep neural networks. However, previous works have primarily addressed this problem from a discrete-state perspective which is not suitable as the number of network layers grows. This paper novelly treats the outputs from layers as states of a continuous process and considers leveraging the state space model (SSM) to design the aggregation of layers in very deep neural networks. Moreover, inspired by its advancements in modeling long sequences, the Selective State Space Models (S6) is employed to design a new module called Selective State Space Model Layer Aggregation (S6LA). This module aims to combine traditional CNN or transformer architectures within a sequential framework, enhancing the representational capabilities of state-of-the-art vision networks. Extensive experiments show that S6LA delivers substantial improvements in both image classification and detection tasks, highlighting the potential of integrating SSMs with contemporary deep learning techniques.

## 1 Introduction

In recent years, the depth of neural network architectures has emerged as a crucial factor influencing performance across various domains, including computer vision, natural language processing, and speech recognition. The network models are capable of capturing increasingly complex features and representations from data as they become deeper, and various methods have emerged to utilize larger numbers of layers to improve performance. For instance, the VGG network (Simonyan & Zisserman, 2015) achieves higher classification accuracy by increasing the number of layers, although its foundation primarily relies on empirical results rather than systematical analysis. Other significant advancements, such as those demonstrated by CNNs (He et al., 2016a; Ren et al., 2016; Tan & Le, 2020) and Transformers (Brown, 2020; Dosovitskiy, 2020; Touvron et al., 2021; Liu et al., 2021; Wang et al., 2022), showcase how deeper architectures can enhance accuracy and generalization.

Growing evidence indicates that strengthening layer interactions can encourage the information flow of a deep neural network. For CNN-based networks, ResNet (He et al., 2016a) employed skip connections, allowing gradients to flow more easily by connecting non-adjacent layers. DenseNet (Huang et al., 2018) extended this concept further by enabling each layer to access all preceding layers within a stage, fostering a rich exchange of information. Later, GLOM (Hinton, 2023) proposed an intensely interactive architecture that incorporates bottom-up, top-down, and same-level connections to effectively represent part-whole hierarchies. Recently, some studies have begun to frame

---

*Authors contributed equally. † Corresponding author.

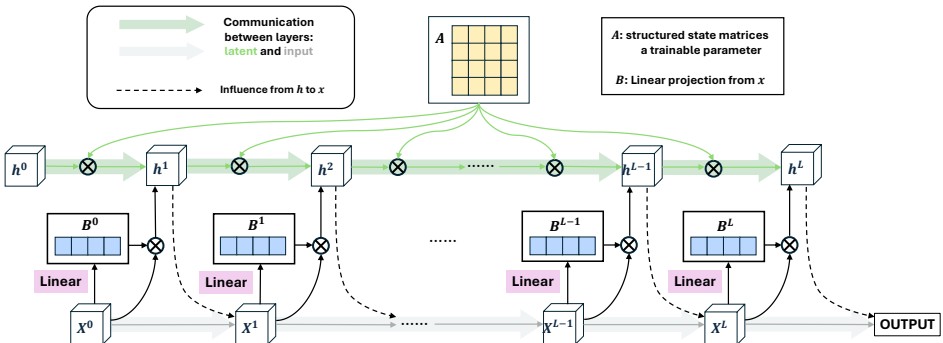

Figure 1: Schematic diagram of a Network with Selective State Space Model Layer Aggregation.

layer interactions with recurrent models and attention mechanisms, such as RLA (Zhao et al., 2021) and MRLA (Fang et al., 2023). All of the above models have been shown by empirical evidence to outperform those without interdependence across layers, and their achievements are obtained by treating the outputs of network layers as discrete states.

However, the perspective of discrete treatment may not be suitable when a neural network is very deep; say, for example, ResNet-152 has 152 layers. Sander et al. (2022) proposed to treat the ResNet as a discretized version of neural ordinary differential equations, i.e. the whole ResNet is considered as a continuous process with the outputs from layers being the corresponding states; see also Liu et al. (2020). Moreover, Queiruga et al. (2020) argued that deep neural network models can learn to represent continuous dynamical systems by embedding them into continuous perspective. This motivates us to conduct layer aggregation among numerous layers of a neural network by alternatively assuming a continuous process to the outputs of layers.

Meanwhile, the State Space Model (SSM), a mathematical framework for continuously updating physical systems, enabled the modeling of dynamic processes and temporal dependencies in deep learning (Gu & Dao, 2023; Liu et al., 2024). Then, Mamba, a selective state space model (Gu & Dao, 2023), proposed selective mechanism and hardware-aware algorithm, which was particularly adept at addressing long sequence modeling challenges. The selection mechanism allows the model to filter out irrelevant information and remember relevant information infinitely.

The significance of layer aggregation in deep models and the popularity of SSMs lead us to propose a novel perspective: layer dynamics in very deep networks can be viewed as a continuous process with long sequence modeling task solvable by selective state space model (S6). By leveraging interactions between layers, outputs from different layers can be treated as sequential data input for an S6, allowing the model to encapsulate a richer representation of the information derived from the original data. By conceptualizing neural networks as state space models, we introduce a novel structure that integrates traditional models into sequential architectures. This approach opens new research avenues that connect traditional statistical methods with contemporary deep learning techniques. Our proposed Selective State Space Model Layer Aggregation (S6LA) effectively harnesses the benefits of layer interactions while incorporating statistical modeling into vision tasks, such as those performed by CNNs and Vision Transformers (ViTs). A schematic of our model is illustrated in Figure 1, with parameters $(\Delta, A, B)$ indicating the influence of $X$ on the implicit latent state $h$. Here $X^{t-1}$ represents the output at the $(t-1)$-th layer, which can be either a hidden layer in a deep CNN or an attention layer in a transformer model.

The main contributions of our work are given below: (1) For a deep neural network, we treat the outputs from layers as states of a continuous process and attempt to leverage the SSM to design the aggregation of layers. To our best knowledge, this is the first time such a perspective has been presented. (2) This leads to a proposed lightweight module, the Selective State Space Model Layer Aggregation (S6LA) module, and it conceptualizes a neural network as a selective state space model (S6), hence solving the layer interactions by the long sequence modelling selective mechanism. (3) Compared with other SOTA convolutional and transformer-based layer aggregation models, S6LA demonstrates superior performance in classification, detection, and instance segmentation tasks.

## 2 RELATED WORK

**State Space Models.** In the realm of state space models, considerable efforts have been directed toward developing statistical theories. These models are characterized by equations that map a 1-dimensional input signal $x(t)$ to an $N$-dimensional latent state $h(t)$, with the details provided in Equation 1. Inspired by continuous state space models in control systems and combined with HiPPO initialization (Gu et al., 2020), LSSL (Gu et al., 2021b) showcased the potential of state space models in addressing long-range dependency problems. However, due to limitations in memory and computation, their adoption was not widespread until the introduction of structured state space models (S4) (Gu et al., 2021a), which proposed normalizing parameters into a diagonal structure. S4 represents a class of recent sequence models in deep learning, broadly related to RNNs, CNNs and classical state space models. Subsequently, Gu & Dao (2023) introduced the selective structured state space model (S6), which builds upon S4 and demonstrates superior performance compared to transformer backbones in various deep learning tasks, including natural language processing and time series forecasting. More recently, VMamba (Liu et al., 2024) was developed, leveraging the S6 model to replace the transformer mechanism and employing a scanning approach to convert images into patch sequences. Additionally, Graph-Mamba (Wang et al., 2024) represented a pioneering effort to enhance long-range context modeling in graph networks by integrating a Mamba block with an input-dependent node selection mechanism. These advancements indicate that state space models have also been successfully applied to complex tasks across various domains.

**Layer Interaction.** The depth of neural network architectures has emerged as a crucial factor influencing performance. And Figure 4 of Appendix illustrates the enhanced performance of deeper neural networks. To effectively address the challenges posed by deeper models, increasing efforts have been directed toward improving layer interactions in both CNN and transformer-based architectures. Some studies (Hu et al., 2018; Woo et al., 2018; Dosovitskiy, 2020) lay much emphasis on amplifying interactions within a layer. DIANet (Huang et al., 2018) employed a parameter-sharing LSTM throughout the network's depth to capture cross-channel relationships by utilizing information from preceding layers. In RLANet (Zhao et al., 2021), a recurrent neural network module was used to iteratively aggregate information from different layers. For attention mechanism, Fang et al. (2023) proposed to strengthen cross-layer interactions by retrieving query-related information from previous layers. Additionally, RealFormer (He et al., 2020) and EA-Transformer (Wang et al., 2021) both incorporated attention scores from previous layers into the current layer, establishing connections through residual attention. However, these methods face significant memory challenges due to the need to retain features from all encoders, especially when dealing with high-dimensional data and they may lack robust theoretical supports.

## 3 PRELIMINARY AND PROBLEM FORMULATION

### 3.1 REVISITING STATE SPACE MODELS

The state space model is defined below, and it maps a 1-dimensional input signal $x(t)$, a continuous process, to an $N$-dimensional latent state $h(t)$, another continuous process:

$$h'(t) = Ah(t) + Bx(t), \tag{1}$$

where $A \in \mathbb{R}^{N \times N}$ and $B \in \mathbb{R}^{N \times 1}$ are the structured state matrix and weight of influence from input to latent state, respectively. We then can obtain the discretization solution of the above equation:

$$h^t = e^{\Delta A}h^{t-1} + \int_{t-1}^{t} e^{A(t-\tau)}Bx(\tau)d\tau. \tag{2}$$

Together with the zero-order hold (ZOH) condition (Karafyllis & Krstic, 2011), i.e. $x(\tau)$ is a constant at intervals $[t-1, t]$ for all integers $t$, we have

$$h^t = e^{\Delta A}h^{t-1} + (\Delta A)^{-1}(\exp(\Delta A) - I) \cdot \Delta B x^t. \tag{3}$$

As a result, the continuous process at Equation 1 can be replaced by the following discrete sequence:

$$h^t = \overline{A}h^{t-1} + \overline{B}x^t \quad \text{with} \quad \overline{A} = \exp(\Delta A) \quad \text{and} \quad \overline{B} = (\Delta A)^{-1}(\exp(\Delta A) - I) \cdot \Delta B. \tag{4}$$

Following Gu & Dao (2023), we refine the approximation of $\overline{B}$ using the first-order Taylor series:

$$\overline{B} = (\Delta A)^{-1}(\exp(\Delta A) - I) \cdot \Delta B \approx (\Delta A)^{-1}(\Delta A) \cdot \Delta B = \Delta B. \tag{5}$$

The formulas $\overline{A} = f_A(\Delta, A)$ and $\overline{B} = f_B(\Delta, A, B)$ are called the discretization rule, where $B = \text{Linear}_N(x)$ is a linear projection of input $x$ into $N$-dimension vector, and $\Delta = \text{Linear}_1(x)$; see Gu et al. (2021a); Gu & Dao (2023) for details.

## 3.2 CNN LAYER AGGREGATION

Consider a neural network, and let $\boldsymbol{X}^{t-1}$ be the output from its $t$th layer. We then can mathematically formulate the layer aggregation at the $t$th layer below,

$$A^t = g^t(\boldsymbol{X}^0, \boldsymbol{X}^1, \cdots, \boldsymbol{X}^{t-2}, \boldsymbol{X}^{t-1}), \quad \boldsymbol{X}^t = f^t(A^{t-1}, \boldsymbol{X}^{t-1}), \tag{6}$$

where $g^t$ is used to summarize the first $t$ layers, $A^t$ is the aggregated information, and $f^t$ produces the new layer output from the last hidden layer and the given aggregation which contains the previous information. The Hierarchical Layer Aggregation proposed (Yu et al., 2018) can be shown to have such similar mechanism which satisfies Equation 6.

This formulation could be generalized to the special case of CNNs. The traditional CNNs do not contain layer aggregation since the layer output only depends on the last layer output, which overlooks the connection between the several previous layers' influence. DenseNet (Huang et al., 2018) perhaps is the first one for the layer aggregation, and its output at $t$th layer can be formulated into

$$\boldsymbol{X}^t = \text{Conv3}^t[\text{Conv1}^t(\text{Concat}(\boldsymbol{X}^0, \boldsymbol{X}^1, \ldots, \boldsymbol{X}^{t-1}))]. \tag{7}$$

Let $A^t = \text{Conv1}^t(\text{Concat}(\boldsymbol{X}^0, \boldsymbol{X}^1, \ldots, \boldsymbol{X}^{t-1}))$ and $\boldsymbol{X}^t = \text{Conv3}^t(A^t)$, and then DenseNet can be rewritten into our framework at Equation 6. RLA (Zhao et al., 2021) considers a more convenient additive form for the layer aggregation, and it has the form of $A^t = \sum_{i=0}^{t-1} \text{Conv1}_i^{t+1}(\boldsymbol{X}^i)$, where the kernel weights of $\text{Conv1}_i^t$ form a partition of the weights in $\text{Conv1}^t$. As a result, a lightweight aggregation can be formed:

$$\boldsymbol{X}^t = \text{Conv3}^t[A^{t-1} + \text{Conv1}_{t-1}^t(\boldsymbol{X}^{t-1})]. \tag{8}$$

Without loss of generality, ResNets (He et al., 2016a;b) can also be treated as a layer aggregation. Specifically, we can treat the update of $\boldsymbol{X}^t = \boldsymbol{X}^{t-1} + f^{t-1}(\boldsymbol{X}^{t-1})$ with applying the update recursively as $A^t = \sum_{i=0}^{t-1} f^i(\boldsymbol{X}^i) + \boldsymbol{X}^0$ and $\boldsymbol{X}^t = A^{t-1} + \boldsymbol{X}^{t-1}$.

## 3.3 ATTENTION LAYERS AGGREGATION

In this section, we show how to generalize the layer aggregation within a transformer. Consider a simple attention layer with general input $\mathbf{X} \in \mathbb{R}^{L \times D}$ and output $\mathbf{O} \in \mathbb{R}^{L \times D}$. Its query $\mathbf{Q}$, key $\mathbf{K}$ and value $\mathbf{V}$ are given by linear projections $\boldsymbol{W}_q \in \mathbb{R}^{D \times D}$, $\boldsymbol{W}_k \in \mathbb{R}^{D \times D}$ and $\boldsymbol{W}_v \in \mathbb{R}^{D \times D}$, i.e. $\mathbf{Q}^T = \boldsymbol{W}_q\mathbf{X}$, $\mathbf{K}^T = \boldsymbol{W}_k\mathbf{X}$ and $\mathbf{V}^T = \boldsymbol{W}_v\mathbf{X}$. As a result, the output $\mathbf{O}$ has the following mathematical formulation:

$$\mathbf{O} = \text{Self-Attention}(\boldsymbol{X}) = \text{softmax}(\frac{\mathbf{Q}\mathbf{K}^T}{\sqrt{D}})\mathbf{V}. \tag{9}$$

Let $\boldsymbol{X}^t \in \mathbb{R}^{L \times D}$ with $1 \leq t \leq T$ be the output from $t$th layer, where $L$ is the number of tokens, $D$ represents the channel of each token, and $T$ is the number of attention layers. A vanilla transformer can then be formulated into:

$$A^t = \boldsymbol{X}^{t-1} + \text{Self-Attention}(\boldsymbol{X}^{t-1}), \quad \boldsymbol{X}^t = A^t + \text{MLP}(\text{Norm}(A^t)). \tag{10}$$

Note that these simple self-attention layers can only capture the connection between the current layer output and the last output; they are supposed to perform better if the information from previous layers can be considered. To this end, we may leverage the idea given by CNN aggregation to concatenate the previous outputs. Specifically, the vanilla transformer at Equation 10 has the form of:

$$\boldsymbol{X}^t = f^t(g^t(\boldsymbol{X}^0, \cdots, \boldsymbol{X}^{t-1})), \tag{11}$$

where $g^t$ is the attention layer, and $f^t$ is the Add & Norm layer for the $t$-th layer. Following Zhao et al. (2021) at Equation 8, we may then use the recurrent mechanism to combine all the outputs given by attention layers, i.e. replacing $A^t = g^t(\boldsymbol{X}^0, \cdots, \boldsymbol{X}^{t-1})$ by $A^t = A^{t-1} + g^{t-1}(\boldsymbol{X}^{t-1})$.

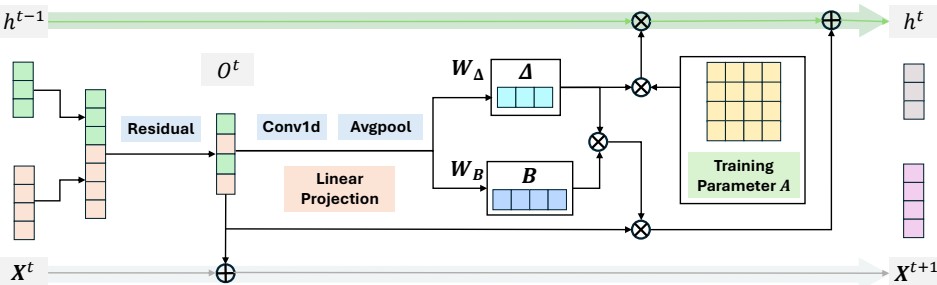

Figure 2: Detailed operations in S6LA module with Convolutional Neural Network. The green arrow shows the hidden state connection, while the grey arrow indicates layers communications.

## 3.4 THE FORMULA OF S6LA

Denote a sequence $\mathbf{X} = \{\boldsymbol{X}^1, \cdots, \boldsymbol{X}^T\}$, where $\boldsymbol{X}^t$ is the output from $t$th layer, say Convolutional layers/blocks or Attention layers, of a deep neural network, and $T$ is the number of layers. In financial statistics, the price of an asset can be treated as a process with discrete time when it is sampled in a low frequency, say weekly data, while it will be treated as a process with continuous time the sampling frequency is high, say one-minute data; see Yuan et al. (2023). Accordingly, we may treat $\mathbf{X}$ as a sequence with discrete time as the number of layers $T$ is small or even moderate, and all existing methods for layer aggregation in the literature follow this way. However, for a very deep neural network, it is more like the scenario of high-frequency data, and hence a continuous process is more suitable for the sequence $\mathbf{X}$ (Sander et al., 2022; Queiruga et al., 2020). This section further conducts layer aggregation by considering state space models in Section 3.1; see Figure 1 for the illustration.

Specifically, we utilize the Mamba model (Gu & Dao, 2023) due to its effectiveness in processing long sequences. This model is based on S6 models and can provide a better interpretation on how to leverage the previous information and then how to store it based on its importance. Moreover, it has been demonstrated to have a better performance than traditional transformers and RNNs. Following its principle, we propose our selective state space model layer aggregation below:

$$h^t = g^t(h^{t-1}, \boldsymbol{X}^t), \quad \boldsymbol{X}^t = f^t(h^{t-1}, \boldsymbol{X}^{t-1}), \tag{12}$$

where $h^t$ is a hidden state similar to $A^t$ in Equation 6, and it represents the recurrently aggregated information up to $(t-1)$th layer. We may consider an additive form, as in Equation 8, for $h^t$. Moreover, $g^t$ is the relation function between the current SSM hidden layer state and previous hidden layer state with input. As a result, similar to Equation 4, the update of $h^t$ can be formulated as:

$$h^t = \overline{A}h^{t-1} + \overline{B}\boldsymbol{X}^t, \quad \boldsymbol{X}^t = f^t(h^{t-1}, \boldsymbol{X}^{t-1}). \tag{13}$$

The choice of function $f^t$ is different for CNNs and Transformer-based models, and they are detailed in the following two subsections; see Figures 6 and 3 for their illustrations, respectively.

## 3.5 APPLICATION TO DEEP CNNS

For CNNs backbones, we adopt ResNet (He et al., 2016a) as the baseline architecture. We propose to concatenate the input at each layer, say $\boldsymbol{X}^t \in \mathbb{R}^{H \times W \times D}$, where $H$ and $W$ represent the height and width, and $D$ indicates the embedding dimension. For each CNN layer, the input to each block in ResNet—comprising a combination of 1D and 3D convolutional operations—is formed by concatenating $\boldsymbol{X}^t$ with the state $h^{t-1} \in \mathbb{R}^{H \times W \times N}$ from the previous layer, where $N$ is the dimension of latent states. This integration effectively incorporates SSM information into the convolutional layers, enhancing the network's capacity to learn complex representations.

For the specific implementation of S6LA in CNNs, we initialize the SSM state $h^0$ using Kaiming normal initialization method (He et al., 2015). This initialization technique is crucial for ensuring effective gradient flow throughout the network, and we will further clarify this point in ablation studies. Next, we employ a selective mechanism to derive two components, the coefficient $B$ for

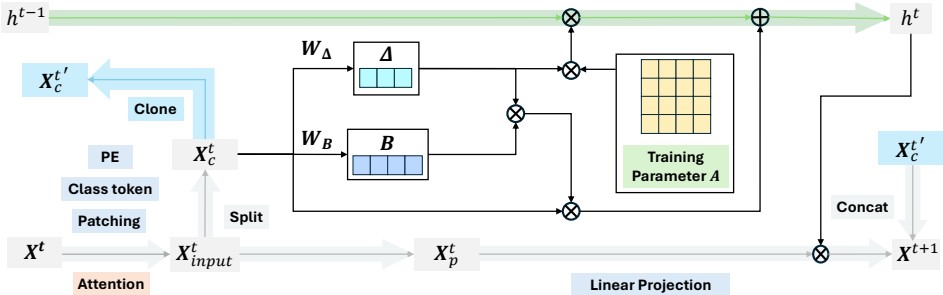

Figure 3: Diagram of the S6LA architecture with Transformer. The green arrow shows the hidden state connection, while the grey arrow indicates communication between layers. The input consists of image patch tokens ($X_p^t$) and a class token ($X_c^t$), processed through positional embedding and attention. The class token is cloned as $X_c^{t'}$, and parameters $W_\Delta$ and $W_B$ update the hidden state ($h^t$). Updated patch tokens are combined with the class token to form the next input ($X^{t+1}$).

input and the interval $\Delta$ as specified in Equation 4. For transition matrix $A$, the initial setting is same as in Mamba models (Gu & Dao, 2023). Then with Equation 4, we can get the next hidden layer $h^t$ based on the last $h^{t-1}$ and $X^t$ for each layer in CNNs.

Utilizing Equation 4, we compute the subsequent latent state $h^t$ based on the previous state $h^{t-1}$ and the input $X^t$ for each layer within the CNN architecture. This methodological framework facilitates improved information flow and retention across layers, thereby enhancing the model's performance. Therefore, the specifics of leveraging our S6LA method with CNNs backbones can be outlined as follows:

**Input Treatment:** We begin by merging the input $X^t$ and the hidden state $h^{t-1}$ through a simple concatenation along the feature dimension. This concatenated representation allows us to generate the output $O^t$ with the CNNs backbone (such as ResNet).

**Latent State Update:** For the input state update, we define $X^{t+1}$ as the sum of $X^t$ and $O^t$. For the hidden state, $h^t$ is derived as a function of these two components, following the formulation provided in Equation 4. The equations are as follows with two trainable parameters $W_\Delta$ and $W_B$ (for the $t-1$ layer):

$$h^t = e^{(\Delta A)} h^{t-1} + \Delta B O^t, \tag{14}$$

where $\Delta = W_\Delta(\text{Conv}(O^t))$, $B = W_B(\text{Conv}(O^t))$.

**Output Computation:** The output component $O^t$ from the input treatment step, contributes to the next input $X^{t+1}$ and the computation is: $X^{t+1} = O^t + X^t$.

### 3.6 APPLICATION TO DEEP ViTs

In our exploration of S6LA implementation in deep ViT backbones, we draw parallels between the integration of the state space model (SSM) state and the mechanisms used in convolutional neural networks (CNNs). However, the methods of combining inputs within transformer blocks differ significantly from those in CNNs. Like the treatment of attention mechanism, we utilize multiplication combination instead of simply concatenating to deal with the input $X^t$ and $h^{t-1}$ in transformer-based models. This approach enhances the interaction between input features and SSM, enabling richer feature representation. Then the next paragraghs gives the specifics of leveraging our S6LA method with ViTs backbones as follows:

**Input Treatment:** We begin by combining the class token and input, alongside the application of positional embeddings. Then following the attention mechanism, $X_{\text{input}}^t \in \mathbb{R}^{(L+1) \times D}$ appeared where $L$ is the number of patches and $D$ is the embedding dimension, and it is split into two components next: image patch tokens $X_p^t \in \mathbb{R}^{L \times D}$ and a class token $X_c^t \in \mathbb{R}^{1 \times D}$.

$$X_{\text{input}}^t = \text{Add\&Norm}(\text{MLP}(\text{Add\&Norm}(\text{Attn}(X^t)))); \quad X_p^t, X_c^t = \text{Split}(X_{\text{input}}^t). \tag{15}$$

Table 1: Comparisons of the Top-1 and Top-5 accuracy on the ImageNet-1K validation set with CNNs. All the results are reproduced by us with 4 GeForce RTX 3090 GPUs with the same parameters. The **bold** fonts denote the best performance.

| Model | Method | Params | FLOPs | Top-1 Acc. | Top-5 Acc. |
|---|---|---|---|---|---|
| ResNet-50 | Vanilla (He et al., 2016a) | 25.6 M | 4.1 B | 76.1 | 92.9 |
| | + SE (Hu et al., 2018) | 28.1 M | 4.1 B | 76.7 | 93.4 |
| | + CBAM (Woo et al., 2018) | 28.1 M | 4.1 B | 77.3 | 93.7 |
| | + $A^2$ (Chen et al., 2018) | 34.6 M | 7.0 B | 77.0 | 93.5 |
| | + AA (Bello et al., 2019) | 27.1 M | 4.5 B | 77.4 | 93.6 |
| | + 1 NL (Wang et al., 2018) | 29.0 M | 4.4 B | 77.2 | 93.5 |
| | + 1 GC (Cao et al., 2019) | 26.9 M | 4.1 B | 77.3 | 93.5 |
| | + all GC (Cao et al., 2019) | 29.4 M | 4.2 B | 77.7 | 93.7 |
| | + ECA (Wang et al., 2020b) | 25.6 M | 4.1 B | 77.4 | 93.6 |
| | + RLA (Zhao et al., 2021) | 25.9 M | 4.3 B | 77.2 | 93.4 |
| | + MRLA (Fang et al., 2023) | 25.7 M | 4.6 B | 77.5 | 93.7 |
| | + S6LA (Ours) | 25.8 M | 4.4 B | **78.0** | **94.2** |
| ResNet-101 | Vanilla (He et al., 2016a) | 44.5 M | 7.8 B | 77.4 | 93.5 |
| | + SE (Hu et al., 2018) | 49.3 M | 7.8 B | 77.6 | 93.9 |
| | + CBAM (Woo et al., 2018) | 49.3 M | 7.9 B | 78.5 | 94.3 |
| | + AA (Bello et al., 2019) | 47.6 M | 8.6 B | 78.7 | 94.4 |
| | + ECA (Wang et al., 2020b) | 44.5 M | 7.8 B | 78.7 | 94.3 |
| | + RLA (Zhao et al., 2021) | 45.0 M | 8.2 B | 78.5 | 94.2 |
| | + MRLA (Fang et al., 2023) | 44.9 M | 8.5 B | 78.7 | 94.4 |
| | + S6LA (Ours) | 45.0 M | 8.3 B | **79.1** | **94.8** |
| ResNet-152 | Vanilla (He et al., 2016a) | 60.2 M | 11.6 B | 78.3 | 94.0 |
| | + SE (Hu et al., 2018) | 66.8 M | 11.6 B | 78.4 | 94.3 |
| | + CBAM (Woo et al., 2018) | 66.8 M | 11.6 B | 78.8 | 94.4 |
| | + AA (Bello et al., 2019) | 66.6 M | 11.9 B | 79.0 | 94.6 |
| | + ECA (Wang et al., 2020b) | 60.2 M | 11.6 B | 78.9 | 94.5 |
| | + RLA (Zhao et al., 2021) | 60.8 M | 12.1 B | 78.8 | 94.4 |
| | + MRLA (Fang et al., 2023) | 60.7 M | 12.4 B | 79.1 | 94.6 |
| | + S6LA (Ours) | 60.8 M | 12.2 B | **79.4** | **94.9** |

The class token plays a crucial role in assessing the correlation between $X^t$ and $h^{t-1}$. Our model setting can effectively bridge the features extracted from the patches with the SSM state by facilitating a better integration into the hidden state since it could be considered as a summary feature of the image in sequential layers.

**Latent State Update:** Given the split class token in last step, the hidden state is updated similar to application in CNNs:

$$h^t = e^{(\Delta A)}h^{t-1} + \Delta B X_c^t, \tag{16}$$

where $\Delta$ and $B$ are calculated from class token with selective mechanism:

$$\Delta = W_\Delta(X_c^t), \quad B = W_B(X_c^t). \tag{17}$$

**Output Computation:** At the same time, the new patch tokens $\widehat{X}_p^t$ are computed as the sum of the previous patch tokens and the product of the previous patch tokens with $h^t$:

$$\widehat{X}_p^t = X_p^t + W X_p^t h^t. \tag{18}$$

Then the next input, $X^{t+1}$, is derived from the concatenation of the updated patch and class tokens:

$$X^{t+1} = \text{Concat}(\widehat{X}_p^t, X_c^t). \tag{19}$$

## 4 EXPERIMENT

This section evaluates our S6LA model in image classification, object detection, and instance segmentation tasks, and provides an ablation study. All models are implemented by the PyTorch toolkit on 4 GeForce RTX 3090 GPUs. More implementation details, and comparisons are provided in Appendix D.

Table 2: Comparisons of the Top-1 and Top-5 accuracy on the ImageNet-1K validation set with vision transformer-based models. All the results are reproduced by us with 4 GeForce RTX 3090 GPUs with the same parameters. The **bold** fonts denote the best performance.

| Backbone | Method | Params | FLOPs | Top-1 | Top-5 |
|---|---|---|---|---|---|
| DeiT | DeiT-Ti (Touvron et al., 2021) | 5.7 M | 1.2 B | 72.6 | 91.1 |
| | + MRLA (Fang et al., 2023) | 5.7 M | 1.4 B | 73.0 | 91.7 |
| | + S6LA (Ours) | 6.1 M | 1.5 B | **73.3** | **92.0** |
| | DeiT-S (Touvron et al., 2021) | 22.1 M | 4.5 B | 79.9 | 95.0 |
| | + MRLA (Fang et al., 2023) | 22.1 M | 4.6 B | 80.7 | 95.3 |
| | + S6LA (Ours) | 23.3 M | 4.8 B | **81.3** | **96.0** |
| | DeiT-B (Touvron et al., 2021) | 86.4 M | 16.8 B | 81.8 | 95.6 |
| | + MRLA (Fang et al., 2023) | 86.5 M | 16.9 B | 82.9 | 96.3 |
| | + S6LA (Ours) | 86.9 M | 17.1 B | **83.3** | **96.5** |
| Swin | Swin-T (Liu et al., 2021) | 28.3 M | 4.5 B | 81.0 | 95.4 |
| | + MRLA (Fang et al., 2023) | 28.9 M | 4.5 B | 80.9 | 95.2 |
| | + S6LA (Ours) | 30.5 M | 4.5 B | **81.5** | **95.6** |
| | Swin-S (Liu et al., 2021) | 49.6 M | 8.7 B | 82.8 | 96.1 |
| | + MRLA (Fang et al., 2023) | 50.9 M | 8.7 B | 82.5 | 96.0 |
| | + S6LA (Ours) | 52.5 M | 8.7 B | **83.3** | **96.5** |
| | Swin-B (Liu et al., 2021) | 87.8 M | 15.4 B | 83.2 | 96.4 |
| | + MRLA (Fang et al., 2023) | 89.8 M | 15.5 B | 82.9 | 96.3 |
| | + S6LA (Ours) | 91.3 M | 15.5 B | **83.5** | **96.6** |
| PVTv2 | PVTv2-B0 (Wang et al., 2022) | 3.4 M | 0.6 B | 70.0 | 89.7 |
| | + MRLA (Fang et al., 2023) | 3.4 M | 0.9 B | 70.6 | 90.0 |
| | + S6LA (Ours) | 3.8 M | 0.6 B | **70.8** | **90.2** |
| | PVTv2-B1 (Wang et al., 2022) | 13.1 M | 2.3 B | 78.3 | 94.3 |
| | + MRLA (Fang et al., 2023) | 13.2 M | 2.4 B | **78.9** | **94.9** |
| | + S6LA (Ours) | 14.5 M | 2.2 B | 78.8 | 94.6 |
| | PVTv2-B2 (Wang et al., 2022) | 25.4 M | 4.0 B | 81.4 | 95.5 |
| | + MRLA (Fang et al., 2023) | 25.5 M | 4.2 B | 81.6 | 95.2 |
| | + S6LA (Ours) | 26.1 M | 4.1 B | **82.3** | **95.9** |

## 4.1 EXPERIMENTS ON IMAGE CLASSIFICATION

**Backbone.** For the dataset, we use Imagenet-1K dataset (Deng et al., 2009) directly. For the CNN backbone, we choose different layers of ResNet (He et al., 2016a). For transformer-based model, DeiT (Touvron et al., 2021), Swin Transformer (Liu et al., 2021) and PVTv2 (Wang et al., 2022) are considered. We compare our S6LA with baselines and other layer aggregation SOTA methods with different backbones alone as the baseline models.

**Experimental settings.** The hyperparameter of state space model channel $N$ shown in Section 3.5 is introduced to control the dimension of feature of $h$ in per S6LA hidden layer module. After comparison of different $N = 16, 32, 64$ for ResNet, we choose 32 as our baseline feature channel, for others we talk about in Section 4.3. In order to compare the baseline models and the models enhanced by S6LA fairly, we use the same data augmentation and training strategies as in their original papers (Zhao et al., 2021; Fang et al., 2023) in all our experiments.

**Main results.** The performance of our main results, along with comparisons to other methods, is presented in Tables 1 and 2. To ensure a fair comparison, all results for the models listed in these tables were reproduced using the same training setup on our workstation. Notably, our model outperforms nearly all baseline models. We specifically compare our S6LA model with other layer interaction methods using ResNets as baselines. The results in Table 1 demonstrate that our S6LA surpasses several layer aggregation methods on CNN backbones, including SENet (Hu et al., 2018), CBAM (Woo et al., 2018), $A^2$-Net (Chen et al., 2018), NL (Wang et al., 2018), ECA-Net (Wang et al., 2020b), RLA-Net (Zhao et al., 2021) and MRLA (Fang et al., 2023). Additionally, we find that our model consistently outperforms them, achieving nearly a 2% improvement in Top-1 accuracy

Table 3: Object detection results of different methods on MS COCO2017. The **bold** fonts denote the best performance.

| Method | Detector | $AP^{bb}$ | $AP^{bb}_{50}$ | $AP^{bb}_{75}$ | $AP^{bb}_{S}$ | $AP^{bb}_{M}$ | $AP^{bb}_{L}$ |
|---|---|---|---|---|---|---|---|
| ResNet-50 (He et al., 2016a) | | 36.4 | 58.2 | 39.2 | 21.8 | 40.0 | 46.2 |
| + SE (Hu et al., 2018) | | 37.7 | 59.1 | 40.9 | 22.9 | 41.9 | 48.2 |
| + ECA (Wang et al., 2020b) | | 38.0 | 60.6 | 40.9 | 23.4 | 42.1 | 48.0 |
| + RLA (Zhao et al., 2021) | | 38.8 | 59.6 | 42.0 | 22.5 | 42.9 | 49.5 |
| + MRLA (Fang et al., 2023) | | 40.1 | 61.3 | **43.8** | 24.0 | 43.9 | 52.2 |
| + S6LA (Ours) | Faster | **40.3** | **61.7** | **43.8** | **24.2** | **44.0** | **52.5** |
| ResNet-101 (He et al., 2016a) | R-CNN | 38.7 | 60.6 | 41.9 | 22.7 | 43.2 | 50.4 |
| + SE (Hu et al., 2018) | | 39.6 | 62.0 | 43.1 | 23.7 | 44.0 | 51.4 |
| + ECA (Wang et al., 2020b) | | 40.3 | 62.9 | 44.0 | 24.5 | 44.7 | 51.3 |
| + RLA (Zhao et al., 2021) | | 41.2 | 61.8 | 44.9 | 23.7 | 45.7 | 53.8 |
| + MRLA (Fang et al., 2023) | | 41.3 | 62.9 | 45.0 | **24.7** | 45.5 | 53.8 |
| + S6LA (Ours) | | **41.7** | **63.0** | **45.2** | 24.6 | **45.6** | **53.9** |
| ResNet-50 (He et al., 2016a) | | 35.6 | 55.5 | 38.2 | 20.0 | 39.6 | 46.8 |
| + SE (Hu et al., 2018) | | 37.1 | 57.2 | 39.9 | 21.2 | 40.7 | 49.3 |
| + ECA (Wang et al., 2020b) | | 37.3 | 57.7 | 39.6 | 21.9 | 41.3 | 48.9 |
| + RLA (Zhao et al., 2021) | | 37.9 | 57.0 | 40.8 | 22.0 | 41.7 | 49.2 |
| + MRLA (Fang et al., 2023) | | 39.1 | 58.6 | **42.0** | 23.6 | **43.3** | 50.8 |
| + S6LA (Ours) | RetinaNet | **39.3** | **59.0** | 41.9 | **23.7** | 42.9 | **51.0** |
| ResNet-101 (He et al., 2016a) | | 37.7 | 57.5 | 40.4 | 21.1 | 42.2 | 49.5 |
| + SE (Hu et al., 2018) | | 38.7 | 59.1 | 41.6 | 22.1 | 43.1 | 50.9 |
| + ECA (Wang et al., 2020b) | | 39.1 | 59.9 | 41.8 | 22.8 | 43.4 | 50.6 |
| + RLA (Zhao et al., 2021) | | 40.3 | 59.8 | 43.5 | 24.2 | 43.8 | 52.7 |
| + MRLA (Fang et al., 2023) | | 41.0 | 60.0 | 43.5 | 24.3 | 44.1 | 52.8 |
| + S6LA (Ours) | | **41.2** | **60.4** | **43.8** | **24.9** | **45.1** | **53.0** |

with only 0.3 B FLOPs compared to vanilla ResNet models. Moreover, in comparison with the latest state-of-the-art method MRLA (Fang et al., 2023), our approach demonstrates fewer FLOPs and higher accuracy. As indicated in Table 2, our S6LA achieves nearly a 1.5% improvement in Top-1 accuracy on vanilla vision transformer-based backbones such as DeiT (Touvron et al., 2021), Swin Transformer (Liu et al., 2021), and PVTv2 (Wang et al., 2022), with only a slight increase in parameters (+0.2 M) and FLOPs (+0.3 B), all within acceptable limits for hardware. Again, when compared to the latest SOTA method MRLA (Fang et al., 2023), our model shows fewer FLOPs and better performance.

## 4.2 EXPERIMENTS ON OBJECT DETECTION AND INSTANCE SEGMENTATION

This subsection validates the transferability and the generalization ability of our model on object detection and segmentation tasks using the three typical detection frameworks: Faster R-CNN (Ren et al., 2016), RetinaNet (Lin et al., 2018) and Mask R-CNN (He et al., 2018).

**Experimental settings.** For the dataset, we choose MS COCO 2017 (Lin et al., 2014) for experiments. All the codes are based on the toolkits of MMDetection (Chen et al., 2019). The hyperparameter of state space model channel $N$ is introduced to control the dimension of feature of $h$ in per S6LA hidden layer module same to the settings in classification tasks.

**Results of object detection and instance segmentation.** For the results of object detection task, Table 9 illustrates the details about AP of bounding box with the notation $AP^{bb}$. It is apparent that the improvements on all metrics are significant. Also compared with the other stronger backbones and detectors, our method outperforms in this task while we only add a little parameters and FLOPs which can be overlooked by the servers. Meanwhile, Table 4 illustrates our S6LA method's improvements about AP of bounding box and mask on all the metrics with Mask R-CNN as the framework. Also similar to the advantages in object detection task, our method balance the parameters and FLOPs with traditional backbones. From the tables' results, it is proved that our S6LA model is feasible.

## 4.3 ABLATION STUDY

**Different variants of S6LA.** Due to resource limitations, we only experiment with the ResNet-50 and DeiT models on the ImageNet dataset. Our investigation considers several factors: (a) the

Table 4: Object detection and instance segmentation results of different methods on MS COCO2017 with Mask R-CNN as a framework. The **bold** fonts denote the best performance.

| Method | Params | $AP^{bb}$ | $AP^{bb}_{50}$ | $AP^{bb}_{75}$ | $AP^m$ | $AP^m_{50}$ | $AP^m_{75}$ |
|---|---|---|---|---|---|---|---|
| ResNet-50 (He et al., 2016a) | 44.2 M | 37.2 | 58.9 | 40.3 | 34.1 | 55.5 | 36.2 |
| + SE (Hu et al., 2018) | 46.7 M | 38.7 | 60.9 | 42.1 | 35.4 | 57.4 | 37.8 |
| + ECA (Wang et al., 2020b) | 44.2 M | 39.0 | 61.3 | 42.1 | 35.6 | 58.1 | 37.7 |
| + 1 NL (Wang et al., 2018) | 46.5 M | 38.0 | 59.8 | 41.0 | 34.7 | 56.7 | 36.6 |
| + GC (Cao et al., 2019) | 46.9 M | 39.4 | 61.6 | 42.4 | 35.7 | 58.4 | 37.6 |
| + RLA (Zhao et al., 2021) | 44.4 M | 39.5 | 60.1 | 43.4 | 35.6 | 56.9 | 38.0 |
| + MRLA (Fang et al., 2023) | 44.4 M | 40.4 | **61.8** | 44.0 | **36.9** | 57.8 | **38.3** |
| + S6LA (Ours) | 44.9 M | **40.6** | 61.5 | **44.2** | 36.7 | **58.3** | **38.3** |
| ResNet-101 (He et al., 2016a) | 63.2 M | 39.4 | 60.9 | 43.3 | 35.9 | 57.7 | 38.4 |
| + SE (Hu et al., 2018) | 67.9 M | 40.7 | 62.5 | 44.3 | 36.8 | 59.3 | 39.2 |
| + ECA (Wang et al., 2020b) | 63.2 M | 41.3 | 63.1 | 44.8 | 37.4 | 59.9 | 39.8 |
| + 1 NL (Wang et al., 2018) | 65.5 M | 40.8 | 63.1 | 44.5 | 37.1 | 59.9 | 39.2 |
| + GC (Cao et al., 2019) | 68.1 M | 41.1 | 63.6 | 45.0 | 37.4 | 60.1 | 39.6 |
| + RLA (Zhao et al., 2021) | 63.6 M | 41.8 | 62.3 | 46.2 | 37.3 | 59.2 | 40.1 |
| + MRLA (Fang et al., 2023) | 63.6 M | 42.5 | **63.3** | 46.1 | 38.1 | 60.3 | 40.6 |
| + S6LA (Ours) | 64.0 M | **42.7** | **63.3** | **46.2** | **38.3** | **60.5** | **41.0** |

influence of $X$ on $h$ (where the opposite is $h$ randomized for each iteration); (b) the hidden state channels set to 16, 32, and 64; (c) the selective mechanism involving the interval $\Delta$ and coefficient $B$; (d) for the Transformer-based method, using simple concatenation instead of multiplication.

From our analysis of the results presented in Tables 5 and 6, several key findings emerge. Firstly, models incorporating our S6LA framework demonstrate superior performance compared to those without it. Notably, using a trainable parameter $h$ ($h$ is influenced by $X$) yields better performance. Secondly, regardless of whether we use ResNet or DeiT, we find that a channel dimension of $N = 32$ yields the best results. Finally, the selective mechanism is crucial for our model; specifically, for the vision Transformer method (in this case, DeiT), the multiplication of $X$ and $h$ outperforms simple concatenation used in CNN backbones.

Table 5: The influence of trainable $h$ and selective mechanism of $\Delta$ and $B$.

| | Model | Params | Top-1 |
|---|---|---|---|
| ResNet | S6LA | 25.8 M | **78.0** |
| | w/o trainable $h$ | 25.8 M | 77.4 |
| DeiT-Ti | S6LA | 6.1 M | **73.3** |
| | w/o trainable $h$ | 6.1 M | 72.5 |
| ResNet | S6LA | 25.8 M | **78.0** |
| | w/o selective | 25.8 M | 77.3 |
| DeiT-Ti | S6LA | 6.1 M | **73.3** |
| | w/o selective | 6.1 M | 72.7 |

Table 6: The influence of latent dimension $N$ and the treatment of DeiT-Ti.

| | Model | Params | Top-1 |
|---|---|---|---|
| ResNet | $N = 16$ | 25.8 M | 77.9 |
| | $N = 32$ | 25.8 M | **78.0** |
| | $N = 64$ | 25.9 M | 77.7 |
| DeiT-Ti | $N = 16$ | 5.9 M | 72.7 |
| | $N = 32$ | 6.1 M | **73.3** |
| | $N = 64$ | 6.3 M | 72.9 |
| DeiT-Ti (S6LA) | | 6.1 M | **73.3** |
| DeiT-Ti (Concatenation) | | 6.1 M | 72.6 |

## 5 CONCLUSION

In conclusion, we have demonstrated an enhanced representation of information derived from the original data by treating outputs from various layers as sequential data inputs to a state space model (SSM). The proposed Selective State Space Layer Aggregation (S6LA) module uniquely combines layer outputs with a continuous perspective, allowing for a more profound understanding of deep models while employing a selective mechanism. Empirical results indicate that the S6LA module significantly benefits classification and detection tasks, showcasing the utility of statistical theory in addressing long sequence modeling challenges. Looking ahead, we aim to optimize our approach by reducing parameters and FLOPs while enhancing accuracy. Additionally, we see potential for integrating further statistical models into computer science applications, suggesting a promising convergence in these fields.

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

## A    THE CORRELATION BETWEEN ACCURACY AND LAYER SIZE

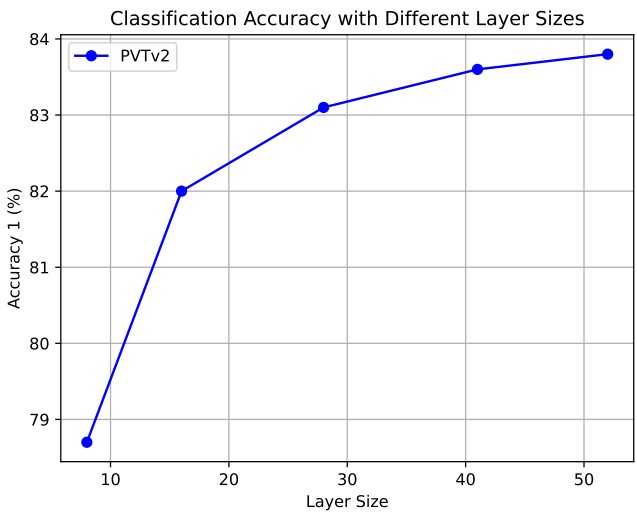

Figure 4: The correlation between accuracy and layer size with model PVT-v2.

Figure 4 illustrates that deeper neural networks can get better performance.

## B    FULL CAPTION OF FIGURES

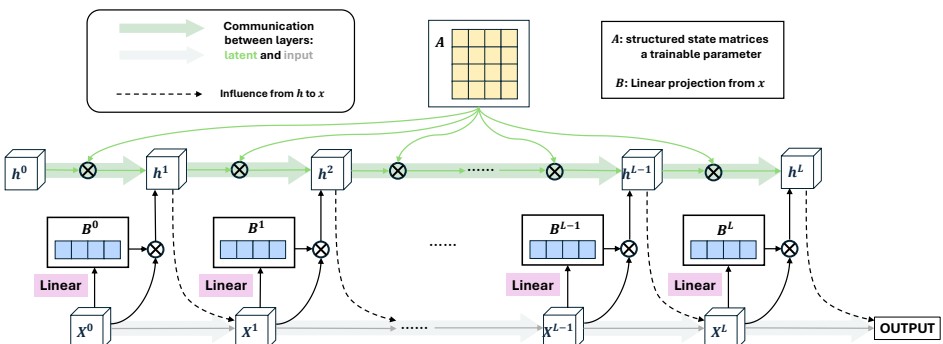

Figure 5: **Schematic Diagram of a Network with Selective State Space Model Layer Aggregation.** The green arrow represents the hidden state connection, while the grey arrow indicates communication between layers. The updated latent layer is derived from the previous latent layer $h^{t-1}$ and the last input layer $\boldsymbol{X}^t$. The output of the $t$-th layer is generated from the input $\boldsymbol{X}^{t-1}$ and the latent layer $h^t$.

## C    PSEUDO CODE

In this section we will propose our pseudo of our method for CNNs and transformers.

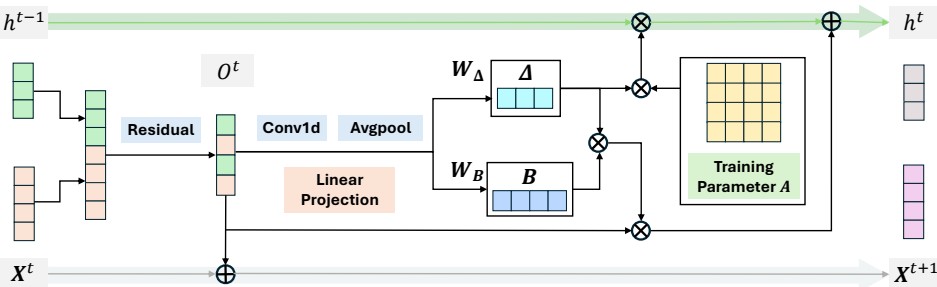

Figure 6: **Detailed Operations in the S6LA Module with Convolutional Neural Network.** The green arrow represents the hidden state connection, while the grey arrow indicates communication between layers. The temporary output $O^t$ is the concatenation of the latent layer $h^{t-1}$ and the input layer $\boldsymbol{X}^t$. After passing through the convolutional layers, the updates for the latent layer $h^t$ and the input layer $\boldsymbol{X}^{t+1}$ are derived from the last layers.

---

**Algorithm 1** S6LA used in CNNs

---

**Input:** Output of the first CNN block $\boldsymbol{X}^1 \in \mathbb{R}^{1 \times H \times W \times D}$, latent state dimension $N$.
**Output:** Output of the $T$-th CNN block $\boldsymbol{X}^T \in \mathbb{R}^{1 \times H \times W \times D}$.
 1: Initial the latent state $h^0$, the trainable structured state matrices $A$;
 2: **for** each $t \in [1, T]$ **do**
 3:    // $h$ influence $\boldsymbol{X}$
 4:    Concatenate the $t-1$-th output $\boldsymbol{X}^{t-1}$ and the $t-1$-th latent state $h^{t-1}$;
 5:    Get the information $\boldsymbol{O}^{t-1}$ concluding the two parts with CNNs and residuals;
 6:    Treat with Conv1d and average pooling to the next step;
 7:    // selective mechanism
 8:    Through linear projection to get the selective parameters $\Delta$ and $B$ from the last step;
 9:    // $X$ influence $h$
10:    Calculate the next output $\boldsymbol{X}^t$ and the update latent state $h^t$ with $\boldsymbol{X}^t = \boldsymbol{O}^{t-1} + \boldsymbol{X}^{t-1}$.
11: **end for**
12: **return** Outputs $\boldsymbol{X}^T$.

---

---

**Algorithm 2** S6LA used in Vision Transformers

---

**Input:** Output of the first CNN block $\boldsymbol{X}^1 \in \mathbb{R}^{1 \times L \times D}$, latent state dimension $N$.
**Output:** Output of the $T$-th CNN block $\boldsymbol{X}^T \in \mathbb{R}^{1 \times L \times D}$.
 1: Initial the latent state $h^0$, the trainable structured state matrices $A$;
 2: **for** each $t \in [1, T]$ **do**
 3:    Combine the class token with the $t-1$-th output $\boldsymbol{X}^{t-1}$ and add positional embeddings;
 4:    Attention mechanism with $\boldsymbol{X}^{t-1}_{\text{input}} = \text{Add\&Norm}(\text{Attn}(\boldsymbol{X}^{t-1}))$ and get $\boldsymbol{X}^{t-1}_{\text{input}} \in \mathbb{R}^{(L+1) \times D}$;
 5:    Split $\boldsymbol{X}^{t-1}_{\text{input}}$ and get the two components $\boldsymbol{X}^{t-1}_p, \boldsymbol{X}^{t-1}_c$;
 6:    // $h$ influence $\boldsymbol{X}$, multiplication not simple concatenation
 7:    The new patch tokens: $\widehat{\boldsymbol{X}}^{t-1}_p = \boldsymbol{X}^{t-1}_p + W \boldsymbol{X}^{t-1}_p h^t$;
 8:    // selective mechanism
 9:    Through linear projection to get the selective parameters $\Delta$ and $B$ from the class token: $\Delta = W_\Delta(\boldsymbol{X}^{t-1}_c), B = W_B(\boldsymbol{X}^{t-1}_c)$;
10:    // $X$ influence $h$
11:    Calculate the next output $\boldsymbol{X}^T$ and the update latent state $h^T$: $h^t = e^{(\Delta A)} h^{t-1} + \Delta B \boldsymbol{X}^{t-1}_c$ and $\boldsymbol{X}^t = \text{Concat}(\widehat{\boldsymbol{X}}^{t-1}_p, \boldsymbol{X}^{t-1}_c)$.
12: **end for**
13: **return** Outputs $\boldsymbol{X}^T$.

---

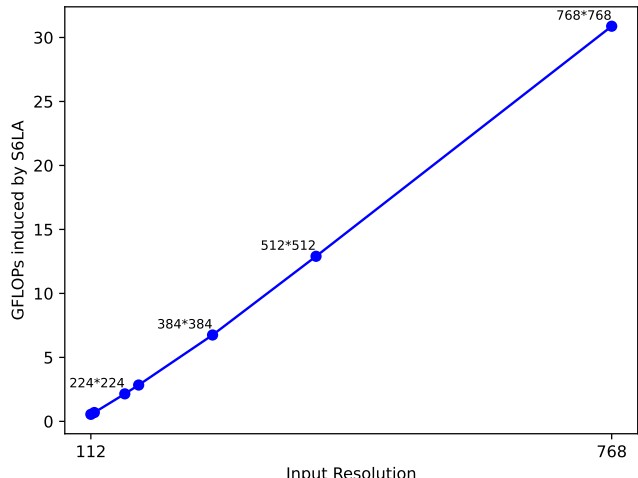

Figure 7: The GFLOPs induced by our method with respect to input resolution with backbone PVT-v2-b1.

## D    EXPERIMENTS

### D.1    IMAGENET CLASSIFICATION

#### D.1.1    IMPLEMENTATION DETAILS

**ResNet**   For training ResNets with our method, we follow exactly the same data augmentation and hyper-parameter settings in original ResNet (He et al., 2016a). Specifically, the input images are randomly cropped to 224 × 224 with random horizontal flipping. The networks are trained from scratch using SGD with momentum of 0.9, weight decay of 1e-4, and a mini-batch size of 256. The models are trained within 100 epochs by setting the initial learning rate to 0.1, which is decreased by a factor of 10 per 30 epochs. Since the data augmentation and training settings used in ResNet are outdated, which are not as powerful as those used by other networks, strengthening layer interactions leads to overfitting on ResNet. Pretraining on a larger dataset and using extra training settings can be an option.

**DeiT, Swin Transformer, PVTv2**   We adopt the same training and augmentation strategy as that in DeiT. All models are trained for 300 epochs using the AdamW optimizer with weight decay of 0.05. We use the cosine learning rate schedule and set the initial learning rate as 0.001 with batch size of 1024. Five epochs are used to gradually warm up the learning rate at the beginning of the training. We apply RandAugment (Cubuk et al., 2020), repeated augmentation (Hoffer et al., 2020), label smoothing (Szegedy et al., 2016) with $\epsilon = 0.1$, Mixup (Zhang, 2017) with 0.8 probability, Cutmix (Yun et al., 2019) with 1.0 probability and random erasing (Zhong et al., 2020) with 0.25 probability. Similarly, our model shares the same probability of the stochastic depth with the MHSA and FFN layers of DeiT/CeiT/PVTv2.

#### D.1.2    MODEL COMPLEXITY WITH RESPECT TO INPUT RESOLUTION

Figure 7 illustrates the FLOPs induced by our model S6LA with respect to input resolution. We use the model PVTv2-b1 as the backbone and then derive the differences under various settings of input resolution. From this, it is apparent that complexity of our method is linear to input resolution.

### D.1.3 COMPARISONS WITH RELEVANT NETWORKS

**Layer-interaction networks**   For the comparison of layer-interaction-based models using CNNs, we first compare our method S6LA with Densenet (Huang et al., 2018), DIANet (Huang et al., 2020), RLA (Zhao et al., 2021) and MRLA (Fang et al., 2023). The comparison on the ImageNet-1K validation set are given in Table 1. From the table, it is obvious that our method outperforms in the classification task and also compared with the similar model size of DenseNet, our model performs better.

**Other relevant networks**   The methods in the last part, they all use the same implemental settings from ours in the training of Imagenet. However for some other models, such as BA-Net (Zhao et al., 2022), adopted other settings. It applied cosine learning schedule and label smoothing in their training process. However, the different settings of our method and their method will give the unfair comparison. Therefore, we adopt the results given in MRLA (Fang et al., 2023) since our method settings are same to this paper. The results are given in Table 8.

Table 7: Performances of layer-interaction-based models on ImageNet-1K validation dataset. The **bold** one denotes the best performance. (The performances of DIANet and DenseNet are from the two papers)

| Model | Params | FLOPs | Top-1 Acc. | Top-5 Acc. |
|---|---|---|---|---|
| ResNet-50 (He et al., 2016a) | 25.6 M | 4.1 B | 76.1 | 92.9 |
| + DIA (Huang et al., 2020) | 28.4 M | - | 77.2 | - |
| + RLA (Zhao et al., 2021) | 25.9 M | 4.5 B | 77.2 | 93.4 |
| + MRLA (Fang et al., 2023) | 25.7 M | 4.6 B | 77.5 | 93.7 |
| + S6LA (Ours) | 25.8 M | 4.4 B | **78.0** | **94.2** |
| ResNet-101 (He et al., 2016a) | 44.5 M | 7.8 B | 77.4 | 93.5 |
| + DIA (Huang et al., 2020) | 47.6 M | - | 78.0 | - |
| + RLA (Zhao et al., 2021) | 45.0 M | 8.4 B | 78.5 | 94.2 |
| + MRLA (Fang et al., 2023) | 44.9 M | 8.5 B | 78.7 | 94.4 |
| + S6LA (Ours) | 45.0 M | 8.3 B | **79.1** | **94.8** |
| DenseNet-161 (k=48) (Huang et al., 2018) | 27.4 M | 7.9 B | 77.7 | 93.8 |
| DenseNet-264 (k=32) (Huang et al., 2018) | 31.8 M | 5.9 B | 77.9 | 93.8 |

Table 8: Performances of BA-Net model on ImageNet-1K validation dataset under our settings. The **bold** one denotes the best performance.

| Model | Params | FLOPs | Top-1 Acc. | Top-5 Acc. |
|---|---|---|---|---|
| BA-Net-50 (Zhao et al., 2022) | 28.7 M | 4.2 B | 77.8 | 93.7 |
| ResNet-50 + S6LA (Ours) | 25.8 M | 4.4 B | **78.0** | **94.2** |

### D.2 OBJECT DETECTION AND INSTANCE SEGMENTATION ON COCO DATASET

**Implementation details**   We adopt the commonly used settings by Hu et al. (2018); Wang et al. (2020b); Cao et al. (2019); Wang et al. (2021); Zhao et al. (2021); Fang et al. (2023), which are same to the default settings in MMDetection toolkit (Chen et al., 2019). For the optimizer, we use SGD with weight decay of 1e-4, momentum of 0.9 and batchsize of 16 for all experiments. The learning rate is 0.02 and is decreased by a factor of 10 after 8 and 11 epochs for within the total 12 epochs. For RetinaNet, we modify the initial learning rate to 0.01 to avoid training problems. For the pretrained model, we use the model trained in ImageNet tasks.

**Results**   For the experiments of detection tasks in the similar settings models, it is illustrated in Table 4 and 9. For almost backbone models, our model performs better and it proves that our model

is also useful in detection tasks. For some other implemental settings such as ResNeXT (Xie et al., 2017) RelationNet++ (Chi et al., 2020), our method also outperforms in object detection results.

Table 9: Object detection results of different methods on MS COCO2017. The **bold** one denotes the best performance.

| Backbone Model | $AP^{bb}$ | $AP^{bb}_{50}$ | $AP^{bb}_{75}$ | $AP^{bb}_{S}$ | $AP^{bb}_{M}$ | $AP^{bb}_{L}$ |
|---|---|---|---|---|---|---|
| ResNet-101 (He et al., 2016a) | 37.7 | 57.5 | 40.4 | 21.2 | 42.2 | 49.5 |
| ResNeXT-101-32x4d (Xie et al., 2017) | 39.9 | 59.6 | 42.7 | 22.3 | **44.2** | 52.5 |
| RelationNet++ (Chi et al., 2020) | 39.4 | 58.2 | 42.5 | - | - | - |
| + S6LA (Ours) | **40.3** | **61.7** | **43.8** | **24.2** | 44.0 | **52.5** |

### D.3 VISUALIZATIONS

To investigate how S6LA contributes to representation learning in convolutional neural networks (CNNs), we utilize the ResNet-50 model (He et al., 2016a) as our backbone. In this study, we visualize the feature maps using score-weighted visual explanations generated by ScoreCAM (Wang et al., 2020a), as illustrated in Figure 8 and 9.

We specifically focus on the final convolutional layer of the ResNet-50 model and our S6LA-enhanced model. This choice is grounded in our observation that the feature maps from the initial three layers of both models exhibit remarkable similarity. In the visualization, the first column presents the original images, the second column displays the ScoreCAM images, and the third column showcases the combination of the original images and their corresponding ScoreCAM. Both two images in our analysis are randomly selected from the ImageNet validation set, ensuring a diverse representation of the data. According to the definition of the CAM method, areas highlighted in warmer colors indicate stronger contributions to the classification decision.

From our visualizations, it is evident that models enhanced with S6LA exhibit larger warm regions, which align more closely with the classification labels. In contrast, the vanilla ResNet-50 model struggles to identify all relevant object areas compared to our method. This disparity suggests that our approach not only improves the localization of important features but also enhances the model's overall classification performance.

The findings presented in the figure provide direct evidence of the efficacy of our method in the classification task. By leveraging S6LA, we can significantly improve the interpretability of CNNs, allowing for better insights into how these models make decisions based on the features they learn. In summary, our results highlight the advantages of incorporating S6LA into standard architectures like ResNet-50, ultimately leading to more robust and accurate classification outcomes.

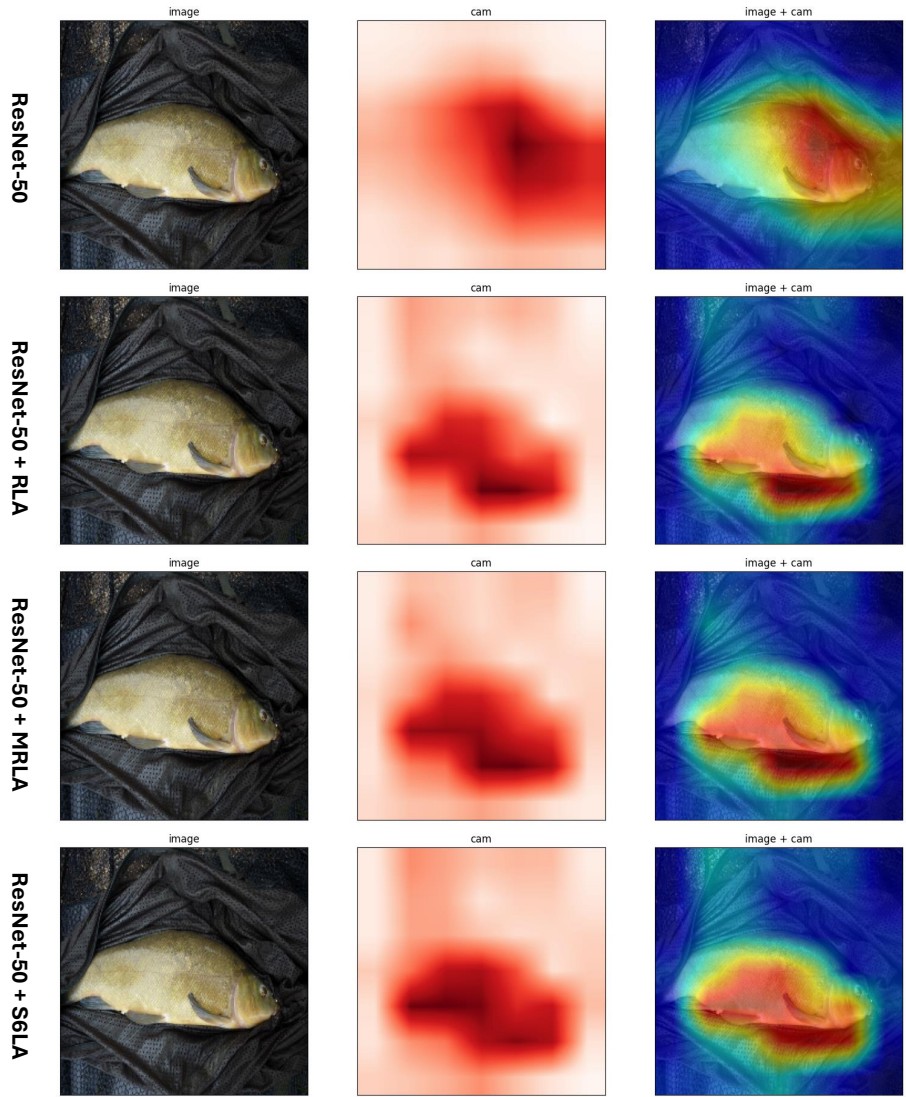

Figure 8: The visualizations of the feature maps extracted from the end convolutional layer of ResNet, RLA, MRLA and our S6LA. The left ones are original images, the middle colomn is the CAM and the right ones are the combinations of left and middle. Compared with others, the red areas of our method are concentrated in the more critical regions of the object (fish) of classification task.

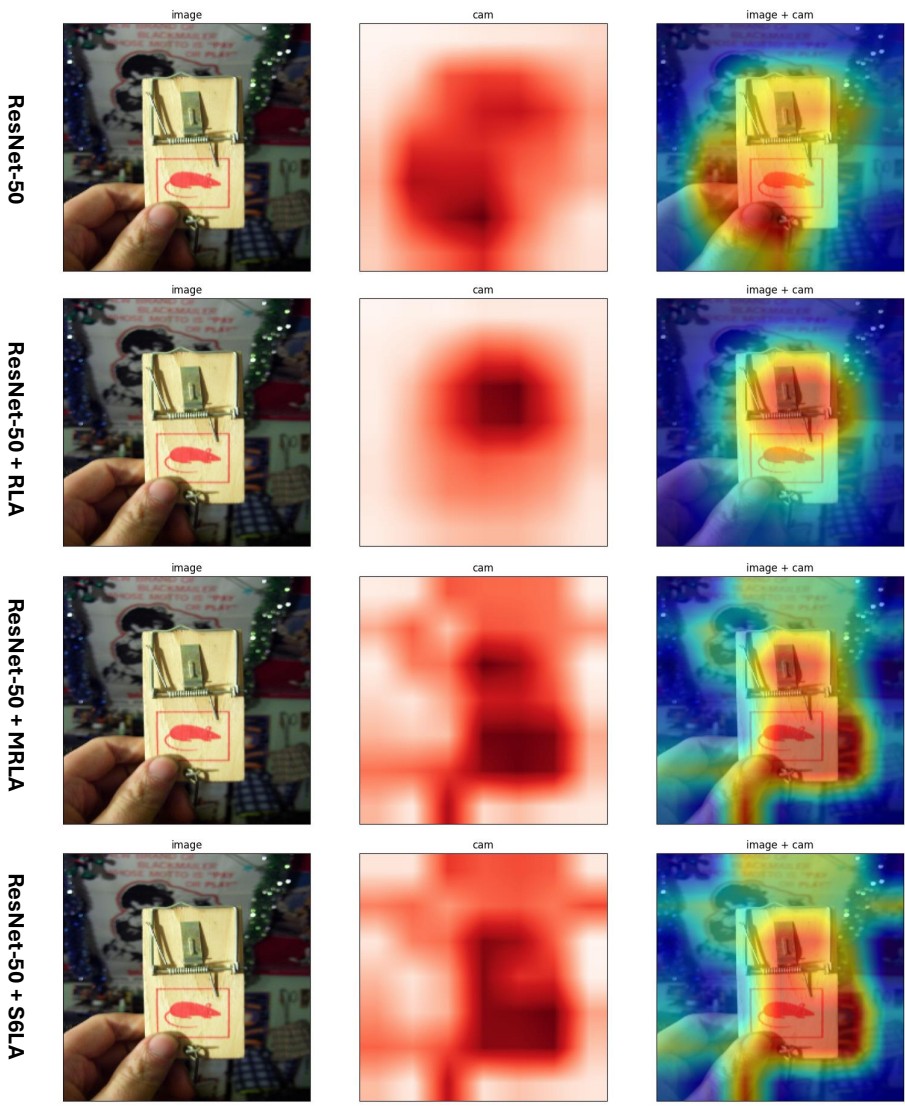

Figure 9: The visualizations of the feature maps extracted from the end convolutional layer of ResNet, RLA, MRLA and our S6LA. The left ones are original images, the middle colomn is the CAM and the right ones are the combinations of left and middle. Compared with others, the red areas of our method are concentrated in the more critical regions of the object of classification task.

