# OpenReview forum: "From Layers to States: A State Space Model Perspective to Deep Neural Network Layer Dynamics"
_ICLR.cc/2025/Conference — ICLR 2025 Poster_

### Official Review · Reviewer_kdSR · 2024-10-27

**Soundness:** 3
**Presentation:** 2
**Contribution:** 3
**Rating:** 6
**Confidence:** 4

**Summary:**

This paper proposes a novel layer aggregation strategy based on the recent S6 block. Two different implementations are conducted for CNN and Transformer architectures. Detailed experiments are delivered on both classification and object detection tasks. The proposed strategy achieves almost the best improvement among current layer aggregation strategies.

**Strengths:**

The proposed layer aggregation based on the S6 block is delicate and achieves significant improvement on various tasks and model architectures. This paper is easy to understand, and the proposed method is easy to follow.

**Weaknesses:**

1. The main problem for me is the different designs between the transformer and CNN. There is a lack of valid explanation here as to why the transformer needs a multiplication strategy while CNN requires an additive one. The only reason given in Lines 301-302 from the input perspective is not convincing enough. Additionally, the performance of CNN with a multiplication format of S6LA (used in the transformer) is missing here.

2. The authors claim that Kaiming normal initialization is crucial for the initial hidden state and will verify this point in the ablation study. However, no relevant experiments are found in the main paper.

3. The impact of S6LA on model efficiency is needed, including detailed inference and training FPS and memory occupation.

4. Some minor errors. The authors may consider replacing the "th" with a subscript, such as the one in Line 164. Besides, from 0 to t-1, there are a total of t layers instead of t+1, as stated in Line 167.

**Questions:**

Could the authors also report the performance improvement on the ADE20K dataset?

---

> ### Author Response · Authors · 2024-11-20
>
> We are deeply grateful to Reviewer kdSR for their positive feedback and constructive critique of our paper. Your insights have been invaluable in guiding our efforts to improve our work.
>
> 1. **Response to Weakness1**
>
>         The main problem for me is the different designs between the transformer and CNN. There is a lack of valid explanation here as to why the transformer needs a multiplication strategy while CNN requires an additive one. The only reason given in Lines 301-302 from the input perspective is
>         not convincing enough. Additionally, the performance of CNN with a multiplication format of S6LA (used in the transformer) is missing here.
>
>     Thank you for your inquiry regarding our choice between concatenation and multiplication in the CNN and Transformer-based methods. We appreciate the chance to clarify our decisions.
>
>     1. **Concatenation in CNN-based S6LA**: In our CNN-based S6LA architecture, we opt for concatenating $X^{t-1}$ and $h^{t-1}$ which is same to the paper [1]. According to the paper by Liu et al. [2], Channel-wise concatenation allows for a more comprehensive integration of features, enabling the model to leverage complementary information from different sources. Unlike multiplication, which may obscure detailed information by compressing feature representations, concatenation retains all critical data, facilitating better learning and representation. Given these points, our choice of concatenation over multiplication is justified, as it allows the S6LA architecture to effectively utilize the detailed information available in the data, enhancing overall model performance.
>
>     2. **Multiplication in Transformer-based S6LA**: For the Transformer-based S6LA architecture, we utilized multiplication to combine the input $X^{t-1}$ and the hidden state $h^{t-1}$, which aligns with traditional attention mechanisms. Intuitively, we believe that our approach is reasonable, and our hypothesis is supported by the results of our experiments. Therefore, we employ multiplication to effectively integrate information from both sources, which enhances the model's ability to concentrate on the most relevant features.
>
>     To illustrate the effectiveness of our chosen methods, we present the following tables comparing the performance of concatenation and multiplication in our CNN and Transformer-based S6LA architectures:
>
>     **CNN-based**:
>
>     | Model | Top-1 | Top-5 |
>     | :---: | :---: | :---: |
>     | ResNet50 + S6LA (Multiplication) | 77.4 | 93.6 |
>     | ResNet50 + S6LA (Concatenation) | 78.0 | 94.2 |
>
>     **Transformer-based**:
>
>     | Model | Top-1 | Top-5 |
>     | :---: | :---: | :---: |
>     | DeiT-Ti + S6LA (Concatenation) | 72.6 | 91.1 |
>     | DeiT-Ti + S6LA (Multiplication) | 73.3 | 92.0 |
>
>     The tables clearly demonstrate that our configurations for both CNN and Transformer-based models achieve superior performance. This underscores the effectiveness of our design choices in enhancing the capabilities of S6LA across various architectures. We appreciate your insightful question, as it allows us to clarify these important aspects of our methodology.
>
> 2. **Response to Weakness2**
>
>         The authors claim that Kaiming normal initialization is crucial for the initial hidden state and will verify this point in the ablation study. However, no relevant experiments are found in the main paper.
>
>     Thank you for your question regarding the experiments involving Kaiming Normal Initialization. We appreciate your interest in this aspect of our methodology, and we will include a table comparing the performance of our model using Kaiming Normal Initialization alongside Gaussian random methods in our revised paper.
>
>     In our experiments, we observed significant differences in performance based on the choice of initialization. For the CNN-based S6LA, when we do not use Kaiming Normal Initialization, we encounter issues with the gradient explosion. This instability highlights the importance of proper initialization in maintaining a stable training process.
>
>     On the other hand, for the Transformer-based S6LA, utilizing Kaiming Normal Initialization resulted in improved performance. This method not only facilitates better convergence but also enhances the overall robustness of the model during training.
>
>     **CNN-based**:
>
>     | Model | Top-1 | Top-5 |
>     | :---: | :---: | :---: |
>     | RseNet50 + S6LA (w/o Kaiming Normal) | N/A | N/A |
>     | ResNet50 + S6LA (Kaiming Normal) | 78.0 | 94.2 |
>
>     **Transformer-based**:
>
>     | Model | Top-1 | Top-5 |
>     | :---: | :---: | :---: |
>     | DeiT-Ti (w/o Kaiming Normal) | 72.5 | 91.1 |
>     | DeiT-Ti (Kaiming Normal) | 73.3 | 92.0 |
>
>     We believe that these findings underscore the critical role of initialization techniques in model performance, and we are committed to providing the above results in our paper. Thank you once again for your valuable feedback; it has been instrumental in helping us refine our work.

---

> ### Author Response · Authors · 2024-11-20
>
> 3. **Response to Weakness3**
>
>         The impact of S6LA on model efficiency is needed, including detailed inference and training FPS and memory occupation.
>
>     We have added the detailed inference and training FPS and memory occupation in the next table and our paper appendix. The training and inference setup is under 4 GeForce RTX 3090 GPUs. The batchsize is 256 for training and 512 for inference. The results are shown below:
>
>     **CNN-based**:
>
>     | Model | Training FPS | Training memory | Inference FPS | Inference memory | Top-1 | Top-5|
>     | :---: | :---: | :---: | :---: | :---: | :---: | :---: |
>     | ResNet50 | 1706 | 7380 M | 2318 | 4568 M | 76.1 | 92.9 |
>     | ResNet50 + RLA | 1333 | 8396 M | 2318 | 5378 M | 77.2 | 93.4 |
>     | ResNet50 + MRLA | 896 | 8501 M | 2334 | 5292 M | 77.5 | 93.7 |
>     | ResNet50 + S6LA | 1095 | 8642 M |  2327 | 5000 M | 78.0 | 94.2 |
>
>     **Transformer-based**:
>
>     | Model | Training FPS | Training memory | Inference FPS | Inference memory | Top-1 | Top-5|
>     | :---: | :---: | :---: | :---: | :---: | :---: | :---: |
>     | DeiT | 1828 | 3381 M | 2949 | 3381 M | 72.6 | 91.1 |
>     | DeiT + MRLA | 1600 | 4218 M | 2961 | 4218 M | 73.0 | 91.7 |
>     | DeiT + S6LA | 1433 | 4447 M | 2960 | 4447 M | 73.3 | 92.0 |
>
>     Based on the experimental results, it is evident that while S6LA incurs additional training costs compared to the standard backbone, the performance improvements are substantial. Moreover, when compared to other layer interaction methods, S6LA's training speed is on par with that of MRLA [1]. We also note that all methods exhibit similar inference speeds, yet S6LA delivers the best performance. In conclusion, we believe that our proposed S6LA method is effective.
>
> 4. **Response to Weakness4**
>
>         Some minor errors. The authors may consider replacing the "th" with a subscript, such as the one in Line 164. Besides, from 0 to t-1, there are a total of t layers instead of t+1, as stated in Line 167.
>
>     Thank you for your careful review of our paper. I sincerely apologize for the errors that were overlooked; we appreciate your diligence in pointing them out and we have corrected them in our revised manuscript.

---

> ### Comment · Reviewer_kdSR · 2024-11-22
>
> Thanks for the detailed response for the authors. Most of my concerns are properly addressed, though for the first question, it's still somewhat intuitive. Therefore, I keep my score of marginally above the acceptance threshold.

---

> > ### Author Response · Authors · 2024-11-24
> >
> > Dear Reviewer kdSR
> >
> > Thank you for your thoughtful feedback and for reviewing the updates to our manuscript. We are pleased to learn that most of your concerns have been addressed.
> >
> > If you have any further comments or suggestions, please do not hesitate to share them. We greatly appreciate your support and consideration.
> >
> > Best regards,
> > Authors of Submission 2351

---

### Official Review · Reviewer_BEKP · 2024-10-29

**Soundness:** 3
**Presentation:** 3
**Contribution:** 3
**Rating:** 8
**Confidence:** 4

**Summary:**

The central idea is to treat various layer outputs as continuous states within a state space model (SSM) and introduce the Selective State Space Model Layer Aggregation (S6LA). The outputs from different layers are fed to the state space model as sequential inputs. This approach aims to enhance the representational power of neural networks by aggregating information across layers in a continuous manner, motivated by advancements in SSM for sequence modeling. Experimental results show that S6LA performs well in comparison to standard convolutional and transformer-based layer aggregation techniques in image classification and object detection tasks.

**Strengths:**

* The approach is novel, application of state space models to layer aggregation in deep neural networks is an interesting concept.
* S6LA demonstrates notable improvements in classification and detection tasks, especially when compared to existing layer aggregation methods.
* The S6LA module is compatible with both CNN and transformer architectures.
* The paper includes extensive experiments on benchmark datasets, showing consistent gains in accuracy.
* The ablation experiments are detailed and well positioned to study the effect different design elements in the approach.

**Weaknesses:**

* Introducing state space models to layer aggregation may increase computational complexity, can the authors add run time comparisons of their and competing methods?
* Authors criticize the RealFormer and EA-transformer approaches for high memory consumption. What is the memory consumption of proposed architecture? It will be good to clarify and compare the memory consumption.
* SSM models can be unstable in training, do authors observe any unstable training behaviour with the proposed SSM based block?

**Questions:**

* How does the computational cost of S6LA compare to simpler aggregation techniques, and are the improvements in accuracy justified by this cost?
* The concatenation vs. multiplication of X and h variables seem to have a lot of impact on final performance. Can the authors motivate why different fusion mechanisms are needed for different architecture families i.e., CNN vs ViTs?

---

> ### Author Response · Authors · 2024-11-20
>
> We are deeply grateful to Reviewer BEKP for their thoughtful feedback and constructive comments on our manuscript. Your detailed review has been really helpful to improve our work. Below, we address each of your points to clarify our methodology and rectify any oversights:
>
> 1. **Response to Weakness1 and Questions1**
>
>         Introducing state space models to layer aggregation may increase computational complexity, can the authors add run time comparisons of their and competing methods?
>
>     Thank you for your question regarding the running time of our models. We appreciate your interest, and we will be sure to include a running time comparison with other competing methods in our revised manuscript.
>
>     The experiments for ImageNet classification were conducted on a workstation with 4 GeForce RTX 3090 GPUs. The batch size is 256 for training and 512 for inference. The average running time of 1 iteration (batch) is shown as follows:
>
>     **CNN-based**:
>
>     | Model | Training Speed | Training memory | Inference Speed | Inference memory | Top-1 | Top-5|
>     | :---: | :---: | :---: | :---: | :---: | :---: | :---: |
>     | ResNet50 | 0.150s | 7380 M | 0.220 s | 4568 M | 76.1 | 92.9 |
>     | ResNet50 + RLA | 0.192s | 8396 M | 0.221 s | 5378 M | 77.2 | 93.4 |
>     | ResNet50 + MRLA | 0.286s | 8501 M | 0.219 s | 5292 M | 77.5 | 93.7 |
>     | ResNet50 + S6LA | 0.235s | 8642 M |  0.220 s | 5000 M | 78.0 | 94.2 |
>
>     **Transformer-based**:
>
>     | Model | Training Speed | Training memory | Inference Speed | Inference memory | Top-1 | Top-5|
>     | :---: | :---: | :---: | :---: | :---: |:---: | :---: |
>     | DeiT | 0.140s | 3381 M | 0.174 s | 3381 M | 72.6 | 91.1 |
>     | DeiT + MRLA | 0.160s | 4218 M | 0.173 s | 4218 M | 73.0 | 91.7 |
>     | DeiT + S6LA | 0.179s | 4447 M | 0.173 s | 4447 M | 73.3 | 92.0 |
>
>     Based on the experimental results, we observe that while S6LA incurs additional training costs compared to the vanilla backbone, the performance improvements are substantial. Furthermore, when compared to other layer interaction methods, S6LA maintains a training speed that is comparable to that of MRLA [1]. Notably, all methods exhibit similar inference speeds; however, S6LA delivers the best overall performance. In summary, we are confident that our proposed S6LA architecture is effective.
>
> 2. **Response to Weakness2**
>
>         Authors criticize the RealFormer and EA-transformer approaches for high memory consumption. What is the memory consumption of proposed architecture? It will be good to clarify and compare the memory consumption.
>
>     In response to your question, first we reference RealFormer and EA-Transformer because both employ layer interaction with attention mechanisms, highlighting the motivation behind our approach.
>
>     While RealFormer utilizes 128 TPU v3 cores (i.e., 64 chips) for BERT-Small/Base/Large and 256 TPU v3 cores (i.e., 128 chips) for BERT-xLarge, we are unable to run these models due to their substantial training costs.
>
>     Although we currently lack empirical data for a direct comparison of memory consumption with these large models, we believe our method can be scaled to larger architectures in the future, as our assumption of treating neural networks as selective state space models is generalizable. We appreciate your understanding and welcome any further discussions on this topic. Thank you for your insightful question!

---

> ### Author Response · Authors · 2024-11-20
>
> 3. **Response to Weakness3**
>
>         SSM models can be unstable in training, do authors observe any unstable training behaviour with the proposed SSM based block?
>
>     Thank you for your insightful questions regarding the stability of our training process. During our implementation, we encountered instability with the SSM. To address this issue, we implemented the following solutions:
>
>     1. **Initialization of Parameters**: To enhance stability in our training, we have chosen to initialize the parameters of $h$ using Kaiming Normal initialization rather than the standard nn.Parameter. This method is specifically designed to maintain a stable variance throughout the layers, which helps to prevent issues that could arise from poor initialization, such as vanishing or exploding gradients.
>
>     2. **Normalization Layer**: In response to observed training instabilities, we have incorporated a normalization layer after our S6LA module. This addition plays a crucial role in ensuring that the training loss remains within a reasonable range. By normalizing the outputs, we facilitate smoother gradient flow, which contributes to a more stable and effective training process.
>
>     3. **Parameter Constraints**: To further ensure stability, we have placed constraints on the absolute value of parameters $A$ and $B$ within our module. These limitations are designed to keep the values of these parameters within a controlled range during training, which reduces the likelihood of instability and enhances the reliability of our model.
>
>     We are committed to continuously improving our methodology and appreciate your understanding as we navigate these challenges. Thank you once again for your thoughtful questions; they are invaluable in helping us refine our work.
>
> 4. **Response to Questions2**
>
>         The concatenation vs. multiplication of X and h variables seem to have a lot of impact on final performance. Can the authors motivate why different fusion mechanisms are needed for different architecture families i.e., CNN vs ViTs?
>
>     Thank you for your question regarding the choice between concatenation and multiplication in our CNN and Transformer-based methods.
>
>     1. **Concatenation in CNN-based S6LA**: In our CNN-based S6LA architecture, we opt for concatenating $X^{t-1}$ and $h^{t-1}$, following the idea in [1]. According to the paper by Liu et al. [2], Channel-wise concatenation in CNNs allows for a more comprehensive integration of features, enabling the model to leverage complementary information from different sources. Unlike multiplication, which may obscure detailed information by compressing feature representations, concatenation retains all critical data, facilitating better learning and representation. Given these points, our choice of concatenation over multiplication is justified, as it allows the S6LA architecture to effectively utilize the detailed information available in the data, enhancing overall model performance.
>
>     2. **Multiplication in Transformer-based S6LA**: For the Transformer-based S6LA architecture, we utilized multiplication to combine the input $X^{t-1}$ and the hidden state $h^{t-1}$, which aligns with traditional attention mechanisms. Intuitively, we believe that our approach is reasonable, and our hypothesis is supported by the results of our experiments. Therefore, we employ multiplication to effectively integrate information from both sources, which enhances the model's ability to concentrate on the most relevant features.
>
>     To illustrate the effectiveness of our chosen methods, we present the following tables comparing the performance of concatenation and multiplication in our CNN and Transformer-based S6LA architectures:
>
>     **CNN-based**:
>
>     | Model | Top-1 | Top-5 |
>     | :---: | :---: | :---: |
>     | ResNet50 + S6LA (Multiplication) | 77.4 | 93.6 |
>     | ResNet50 + S6LA (Concatenation) | 78.0 | 94.2 |
>
>     **Transformer-based**:
>
>     | Model | Top-1 | Top-5 |
>     | :---: | :---: | :---: |
>     | DeiT-Ti + S6LA (Concatenation) | 72.6 | 91.1 |
>     | DeiT-Ti + S6LA (Multiplication) | 73.3 | 92.0 |
>
>     The tables clearly show that our configurations for both the CNN and Transformer-based models achieve superior performance. This highlights the effectiveness of our design choices in enhancing the capabilities of S6LA across different architectures. We appreciate your insightful question, which has provided us with the opportunity to clarify these important aspects of our methodology.
>
> We are committed to continuously improving our methodology and appreciate your understanding as we navigate these challenges. Thank you once again for your thoughtful questions; they are invaluable in helping us refine our work.

---

> > ### Comment · Reviewer_BEKP · 2024-11-27
> >
> > I would like to thank the authors for their responses. I am happy to maintain the inital score.

---

> > > ### Author Response · Authors · 2024-11-28
> > >
> > > Dear Reviewer BEKP,
> > >
> > > Thank you for your kind words and for maintaining the initial score. We appreciate your support and are glad to hear that our responses met your expectations.
> > >
> > > Best regards,
> > > Authors

---

### Official Review · Reviewer_1DRX · 2024-10-29

**Soundness:** 3
**Presentation:** 3
**Contribution:** 3
**Rating:** 8
**Confidence:** 3

**Summary:**

The paper proposes a novel layer aggregation method S6LA that treats the outputs from layers as states of a continuous process and utilizes selective state space models architecture for layer aggregation. The paper reformulates two different layer aggregation processes with the proposed S6LA for ConvNet and Transformer-based backbones. The experiments are conducted on two tasks image classification
and detection on ImageNet, MS COCO 2017 datasets.

**Strengths:**

1. The paper is well-written and well-organized.
2. The idea of utilizing a selective state space-based model for seamless layer aggregation with continuous layer output is novel and interesting. The different integration formulation using the proposed S6LA for different backbones(ConvNet and Tranformer) is reasonable.
3. The quantitative results in the experiment sections show the proposed method's effectiveness over baseline methods.

**Weaknesses:**

1. The pipeline figure (Figure 3) for the Transformer-based approach can be improved so it will be easier for the reader to follow. For example, the naming labels in Figure 3 should be more aligned with what's in the paper along with some pipeline walkthrough captions.
2. Selective State Space models are known for slow training due to their nature of backpropagation through time (even with parallel scans). Will this affect the transformer and ConvNet backbone's parallelization training compared to the baseline methods that are Transformer or ConvNet-based?

**Questions:**

I have a question about transformer-based S6LA integration.

In the paper, equations 15, 18, 19, and Figure 3, $X^{t−1}_{input}$ go through the attention block, add & norm, and split into two parts: the cls embedding goes through the S6LA, image embeddings add with the product of itself with hidden state, and the final output is the concatenated cls embedding and image embedding.

1. Can you provide a clarification that with the S6LA integration, are the MLP blocks excluded from the original Transformer block?

2. If yes, can you clarify the reason for excluding?

3. If no, how it fits into the pipeline detailed in equations 15, 18, 19, and Figure 3?

4. (I am assuming Linear Projection in Figure 3 means MLP block in the Not Exlcuded case) Since in the traditional transformer-based methods,  after attention and add & norm, cls embedding is also fed into the MLP layer along with image embeddings. From Figure 3, the proposed method seems to exclude cls embedding from MLP layers. Can you clarify the reason and potential impact on layer aggregation if this is the case?

---

> ### Author Response · Authors · 2024-11-20
>
> We are grateful to Reviewer 1DRX for your constructive feedback and positive comments on our paper. We appreciate the time and effort you've dedicated to reviewing our work. Please find our responses to your questions below:
>
> 1. **Response to Weakness1**
>
>         The pipeline figure (Figure 3) for the Transformer-based approach can be improved so it will be easier for the reader to follow. For example, the naming labels in Figure 3 should be more aligned with what's in the paper along with some pipeline walkthrough captions.
>
>     Thank you for your valuable suggestions regarding our pipeline figures. We recognize that the current representations are somewhat simplistic and may cause confusion for readers. To enhance clarity and improve understanding, we have revised the captions to include more detailed walkthroughs and introduce the complete process of our model in the transformer. Your feedback is invaluable for improving the quality and clarity of our work. We are committed to making necessary adjustments to ensure our figures are more comprehensible.
>
> 2. **Response to Weakness2**
>
>         Selective State Space models are known for slow training due to their nature of backpropagation through time (even with parallel scans). Will this affect the transformer and ConvNet backbone's parallelization training compared to the baseline methods that are Transformer or ConvNet-based?
>
>     Thank you for your questions about the running time of our method compared with the baseline methods of Transformer and CNN-based. The experiments for ImageNet classification were conducted on a workstation with 4 GeForce RTX 3090 GPUs. The batch size is 256 for training and 512 for inference. The average running time of 1 iteration (batch) is shown as follows:
>
>     **CNN-based**:
>
>     | Model | Training Speed | Training memory | Inference Speed | Inference memory | Top-1 | Top-5|
>     | :---: | :---: | :---: | :---: | :---: | :---: | :---: |
>     | ResNet50 | 0.150s | 7380 M | 0.220 s | 4568 M | 76.1 | 92.9 |
>     | ResNet50 + RLA | 0.192s | 8396 M | 0.221 s | 5378 M | 77.2 | 93.4 |
>     | ResNet50 + MRLA | 0.286s | 8501 M | 0.219 s | 5292 M | 77.5 | 93.7 |
>     | RseNet50 + S6LA | 0.235s | 8642 M |  0.220 s | 5000 M | 78.0 | 94.2 |
>
>     **Transformer-based**:
>
>     | Model | Training Speed | Training memory | Inference Speed | Inference memory | Top-1 | Top-5|
>     | :---: | :---: | :---: | :---: | :---: |:---: | :---: |
>     | DeiT | 0.140s | 3381 M | 0.174 s | 3381 M | 72.6 | 91.1 |
>     | DeiT + MRLA | 0.160s | 4218 M | 0.173 s | 4218 M | 73.0 | 91.7 |
>     | DeiT + S6LA | 0.179s | 4447 M | 0.173 s | 4447 M | 73.3 | 92.0 |
>
>     Based on the above experimental results, we can see that although S6LA introduces additional training cost compared to the vanilla backbone, the performance improvement is significant. Additionally, compared to other layer interaction methods, the training speed of S6LA is comparable to that of MRLA [1]. Furthermore, we observe that all methods have similar inference speeds, but S6LA achieves the best performance. In summary, we believe that our proposed S6LA is effective.

---

> > ### Author Response · Authors · 2024-11-20
> >
> > 3. **Response to Questions**
> >
> >         I have a question about transformer-based S6LA integration.
> >
> >         In the paper, equations 15, 18, 19, and Figure 3,
> >         go through the attention block, add & norm, and split into two parts: the cls embedding goes through the S6LA, image embeddings add with the product of itself with hidden state, and the final output is the concatenated cls embedding and image embedding.
> >
> >         Can you provide a clarification that with the S6LA integration, are the MLP blocks excluded from the original Transformer block?
> >
> >         If yes, can you clarify the reason for excluding?
> >
> >         If no, how it fits into the pipeline detailed in equations 15, 18, 19, and Figure 3?
> >
> >         (I am assuming Linear Projection in Figure 3 means MLP block in the Not Exlcuded case) Since in the traditional transformer-based methods, after attention and add & norm, cls embedding is also fed into the MLP layer along with image embeddings. From Figure 3, the proposed method seems to exclude cls embedding from MLP layers. Can you clarify the reason and potential impact on layer aggregation if this is the case?
> >
> >     Thank you for your in-depth interest regarding the MLP blocks in the transformer-based integration of S6LA. We sincerely apologize for overlooking MLP blocks from both Equation 15 and Figure 3.
> >
> >     To clarify, we incorporate the MLP blocks after the attention block in our architecture. Therefore, Equation 15 should be modified to accurately reflect this structure. We have modified Equation 15 and Figure 3 in our revised munuscript.
> >
> > We appreciate your careful review of our work, as it helps us ensure the accuracy and clarity of our manuscript.
> >
> > Reference:
> >
> > [1] Yanwen Fang, Yuxi Cai, Jintai Chen, Jingyu Zhao, Guangjian Tian, and Guodong Li. Cross-layer retrospective retrieving via layer attention. ICLR, 2023.

---

> ### Comment · Reviewer_1DRX · 2024-11-20
>
> I appreciate the author's response. Most of my concerns have been addressed. The updated Equation 15 and Figure 3 significantly enhance clarity. I will keep my current rating of marginal acceptance and continue to pay attention to other reviews and ongoing discussions.

---

> > ### Author Response · Authors · 2024-11-24
> >
> > Dear Reviewer 1DRX,
> >
> > Thank you for your thoughtful feedback and for taking the time to review the updates to our manuscript. We are glad to hear that most of your concerns have been addressed.
> >
> > If you have any additional comments or suggestions, please feel free to share them. Thank you once again for your support and consideration.
> >
> > Best regards,
> > Authors of Submission2351

---

> > > ### Comment · Reviewer_1DRX · 2024-12-02
> > >
> > > I appreciate the author's new discussion responses and updated version of the paper. I have reread the new version of the paper. I am willing to raise my score because:
> > > 1. Most of reviewers' major concerns are about some paper clarity. The authors addressed those concerns through rebuttal discussion and especially with the updated manuscript. The paper is now with a much better clarity and the logic is easy to follow as a reader.
> > > 2. Some concerns are around the question of amba architecture, and the authors addressed those concerns with quantitative results and explanations.
> > > 3. The method integration is novel and the experimental results are well.

---

> > > > ### Author Response · Authors · 2024-12-03
> > > >
> > > > Dear Reviewer 1DRX,
> > > >
> > > > Thank you very much for your thoughtful feedback and for taking the time to reread the updated version of our paper. We deeply appreciate your willingness to reconsider your score and your kind words about the improved clarity and logical flow of the manuscript.
> > > >
> > > > Best,
> > > > Authors

---

### Official Review · Reviewer_nBY3 · 2024-11-04

**Soundness:** 2
**Presentation:** 2
**Contribution:** 2
**Rating:** 5
**Confidence:** 3

**Summary:**

The paper presents a new module named Selective State Space Model Layer Aggregation (S6LA), designed to integrate traditional CNN or transformer architectures into a sequential framework for improving the representational power of advanced vision networks. The paper shows results on image classification and detection and segmentation benchmarks.

**Strengths:**

Overall the paper seems to be structured.

**Weaknesses:**

The main issue of the paper is that it is not written with enough clarity and often sidetracks from the main points making it hard to understand the main point. Unfortunately, the lack of clarity in the paper makes it hard to judge the work itself. The idea pitched forward in the paper might be interesting but in its current form, it's remotely comprehensible. I would suggest that the author(s) go over the paper and rewrite it in as simple and clear as possible. Also, please go over the mathematical notation and polish it more. Moreover, please pay attention to the citations to back up the claims. For example, lines 230-234, does not contain any citation or mathematical grounds for the claims made there. Similarly, line 95-96 is cryptic. Moreover, the captions of the figures should also indicate what is happening in the figure (this goes for all the figures included in the paper). I believe that once the paper writing is majorly polished and refined, it may have a higher chance of getting through.

**Questions:**

What kind of results does the table 4 contains? Does it use Mask RCNN 1x schedule, 3x schedule (from implementation details it seems like it is 1x schedule but then why is 3x schedule not included)?
In fig. 6 and fig 7, Can the author(s) please comment or write in the caption how the qualitative results for the proposed technique are better than the competition?

---

> ### Author Response · Authors · 2024-11-20
>
> We are deeply grateful to Reviewer nBY3 for the crucial feedback and constructive critique of our paper. Your insights have been invaluable in guiding our efforts to improve our work.
>
> 1. **Response to Weakness**
>
>         The main issue of the paper is that it is not written with enough clarity and often sidetracks from the main points making it hard to understand the main point. Unfortunately, the lack of clarity in the paper makes it hard to judge the work. The idea pitched forward in the paper might be interesting but in its current form, it's remotely comprehensible. I would suggest that the author(s) go over the paper and rewrite it in as simple and clear as possible. Moreover, please pay attention to the citations to back up the claims. For example, lines 230-234, does not contain any citation or mathematical grounds for the claims made there. Similarly, line 95-96 is cryptic. Moreover, the captions of the figures should also indicate what is happening in the figure (this goes for all the figures included in the paper). I believe that once the paper writing is majorly polished and refined, it may have a higher chance of getting through.
>
>     We appreciate the opportunity to clarify the main idea of our paper: we treat the outputs from layers as states of a continuous process and attempt to leverage the SSM to design the aggregation of layers. To the best of our knowledge, this is the first time such a perspective has been presented. To better articulate our ideas, we begin with the below essential **background** in our manuscript:
>
>     1. **Importance of Network Depth:** [1] and [2] point out how the depth of neural network architectures has become a critical factor influencing performance across various domains. This has led to the emergence of very deep neural networks in recent research.
>
>     2. **Layer Interaction:** Layer interaction means reusing information from previous layers to better extract features at the current layer. For example, ResNet [1] employed skip connections, allowing gradients to flow more easily by connecting non-adjacent layers. Therefore, drawing on existing literature, we affirm that layer interaction is beneficial for performance.
>
>     Under these backgrounds, the significance of layer aggregation in deep models and the popularity of SSMs lead us to propose a novel perspective that layer dynamics in very deep networks can be viewed as a continuous process with long sequence modelling tasks solvable by a selective state space model. With this assumption, the **contributions** of our study are three-fold:
>
>     1. **Layers treatment from continuous perspective:** Unlike traditional approaches[5,6], which treat the outputs of network layers as discrete states, **for the first time**, we propose the application of state space models, mathematical frameworks used for **continuous** physical systems, to facilitate layer interactions in very deep neural networks. By treating outputs from different layers as sequential data inputs for the state space model, we can encapsulate a richer representation of the information derived from the original data.
>
>     2. **Implementation of S6LA for better layer interaction:** We propose a lightweight module, the Selective State Space Model Layer Aggregation module **S6LA**, to conceptualize a neural network as a selective state space model (S6). In S6LA, we divide the layer interaction into two steps. In the first latent state update step, we update the hidden state with the feature from the previous layer according the mechanism in S6. In the second output computation step, we calculate the output of current layer, applying concatenation for CNNs and multiplication for Transformer-based models with the updated hidden state.
>
>     3. **Comprehensive experimental validation:** In comparison with other SOTA convolutional and transformer-based layer aggregation models, our method demonstrates superior performance in classification, detection, and instance segmentation tasks.
>
>     Other specific questions:
>
>     1. In response to the comments regarding lines 230-234, We acknowledge that several studies indicate high-frequency data can be approximated as continuous processes, and we will incorporate relevant citations in our revised manuscript, including references [3] and [4].
>
>     2. In response to the comments for lines 95-96 Above, we provide a more detailed explanation of our contributions in the above response as well as in our revised manuscript.
>
>     3. In response to feedback regarding our mathematical notation, we carefully reviewed all equations and notations in our manuscript and revised Equation 15 for improved clarity.
>
>     We appreciate your feedback; we believe the above revisions significantly improve the quality of our work. In addition, we would be more than happy if you could kindly provide further suggestions or specific concerns about our paper. This would greatly assist us in offering a more focused and thorough response.

---

> > ### Author Response · Authors · 2024-11-20
> >
> > 2. **Response to Question1**
> >
> >         What kind of results does the table 4 contains? Does it use Mask RCNN 1x schedule, 3x schedule (from implementation details it seems like it is 1x schedule but then why is 3x schedule not included)?
> >
> >     In our manuscript, we utilized the 1x schedule (12 epochs) to maintain consistency with prior studies, including [5,6]. We appreciate your suggestion to explore performance across different schedules. Currently, our experiments are conducted using MMDetection, which supports both 1x and 2x schedules. To address your concern, we have incorporated experiments using the 2x schedule (24 epochs).
> >
> >     We recognize the potential benefits of exploring the 2x schedule (training for 24 epochs) to evaluate the performance of both models: ResNet50 and our modified ResNet50\textsubscript{s6}. Specifically, The results for ResNet50 and our method are presented in the following table, both trained on four GeForce RTX 3090 GPUs:
> >
> >
> >     | Method (1x schedule) | $AP^{bb}$ | $AP^{bb}_{50}$ | $AP^{bb}_{75}$ | $AP^{m}$ | $AP^{m}_{50}$ | $AP^{m}_{75}$ |
> >     | :---: | :---: | :---: | :---: | :---: | :---: | :---: |
> >     | ResNet50 | 0.372 | 0.589 | 0.403 | 0.341 | 0.555 | 0.362 |
> >     | +S6LA (Ours) | 0.406 | 0.615 | 0.442 | 0.367 | 0.583 | 0.383 |
> >
> >     | Method (2x schedule) | $AP^{bb}$ | $AP^{bb}_{50}$ | $AP^{bb}_{75}$ | $AP^{m}$ | $AP^{m}_{50}$ | $AP^{m}_{75}$ |
> >     | :---: | :---: | :---: | :---: | :---: | :---: | :---: |
> >     | ResNet50 | 0.378 | 0.580 | 0.411 | 0.345 | 0.549 | 0.368 |
> >     | +S6LA (Ours) | 0.415 | 0.625 | 0.451 | 0.374 | 0.593 | 0.398 |
> >
> >
> >     From the results, we observe that your suggestion to train with the 2x schedule indeed yields valuable insights and enhances the robustness of our results. Specifically, we would like to emphasize that our S6LA significantly enhances performance compared to the vanilla backbone when using the 2x schedule; for instance, we noticed an improvement above 0.04 on $AP^{bb}$, $AP^{bb}_{50}$, and $AP^{bb}_{75}$. We appreciate your constructive suggestions, and we are confident that these efforts will further strengthen the validity of our findings.
> >
> > 2. **Response to Question2**
> >
> >         Moreover, the captions of the figures should also indicate what is happening in the figure (this goes for all the figures included in the paper).
> >         In fig. 6 and fig 7, Can the author(s) please comment or write in the caption how the qualitative results for the proposed technique are better than the competition?
> >
> >     Thank you for your question regarding the captions of Figures. We have made the necessary updates. The reason our model outperforms others is that the red areas of our method are concentrated in the more critical regions of the object of classification task. We have also reviewed all the figures and updated the captions for Figures 2 and 3 to clarify what is happening in the figure.
> >
> > We have updated the above revisions in our manuscript. We hope that our revised responses adequately address the concerns raised by Reviewer nBY3. We sincerely appreciate the opportunity to enhance our manuscript based on your valuable feedback and look forward to any further discussion you may have with us.
> >
> > Reference:
> >
> > [1] Kaiming He, Xiangyu Zhang, Shaoqing Ren, and Jian Sun. Deep residual learning for image recognition. In Proceedings of the IEEE conference on computer vision and pattern recognition, pp. 770–778, 2016a.
> >
> > [2] Alejandro F Queiruga, N Benjamin Erichson, Dane Taylor, and Michael W Mahoney. Continuous-
> > in-depth neural networks. arXiv preprint arXiv:2008.02389, 2020.
> >
> > [3] Todorov V. Estimation of continuous-time stochastic volatility models with jumps using high-frequency data[J]. Journal of Econometrics, 2009, 148(2): 131-148.
> >
> > [4] Khorramabadi H, Gray P R. High-frequency CMOS continuous-time filters[J]. IEEE Journal of Solid-State Circuits, 1984, 19(6): 939-948.
> >
> > [5] Yanwen Fang, Yuxi Cai, Jintai Chen, Jingyu Zhao, Guangjian Tian, and Guodong Li. Cross-layer retrospective retrieving via layer attention. arXiv preprint arXiv:2302.03985, 2023.
> >
> > [6] Jingyu Zhao, Yanwen Fang, and Guodong Li. Recurrence along depth: Deep convolutional neural networks with recurrent layer aggregation. Advances in Neural Information Processing Systems, 34:10627–10640, 2021.

---

> ### Author Response · Authors · 2024-11-24
> **Follow-Up on Manuscript Review**
>
> Dear Reviewer nBY3,
>
> I hope this message finds you well. As the deadline for the rebuttal approaches in two days, we would greatly appreciate any feedback or insights you may have.
>
> We understand that reviewing can be time-consuming and we sincerely appreciate your efforts. Your input is invaluable to us, and we are eager to address any comments or questions to improve our work.
>
> Thank you again for your time and consideration. We look forward to hearing from you soon.
>
> Best regards,
> Authors of Submission2351

---

> > ### Comment · Reviewer_nBY3 · 2024-11-24
> >
> > Dear Author(s),
> >
> > I hope you are doing well. Thank you for the gentle reminder. I am currently in the process of gathering and writing down my response and further feedback on the revised manuscript. I will post it today for sure. I will be quick in responding to further discussion in this quite limited time of two days. I appreciate your patience.  I look forward to a constructive discussion as well.
> >
> > Best regards

---

> > > ### Comment · Reviewer_nBY3 · 2024-11-24
> > >
> > > Dear Author(s),
> > >
> > > Thank you for your patience. I have reviewed the revised manuscript and appreciate the authors’ efforts in improving the draft.  While the second version shows notable enhancements compared to the initial submission, there are still significant issues affecting its overall quality and clarity.  I will outline my specific concerns, addressing them sequentially from the beginning to the end of the manuscript. If I have misunderstood any aspect, please do not hesitate to correct me.
> > >
> > > 1. In the abstract, it is mentioned that "This paper novelly treats the outputs from layers as states of a continuous process". This is confusing because continuous processes exist in real life as compared to a finite model with finite weights and biases. The term is used again in lines 069-070, 081, 094, 153, 235. One can imagine that this term will be defined or explained in the rest of the paper but it is not the case.
> > > 2. The motivation of the paper is not clear and seems to be forced specially due to overly dense, abstract and cryptic use of language such as in line 042 "strengthening layer interactions", lines 053-070 (what is a "continuous perspective"?), lines 097-098.
> > > 3. Inconsistent and contradictory use of mathematical notation. e.g. on line 090 it is never defined what exactly is $x$ itself and $t$?
> > >  4. On Line 161 it is mentioned that
> > > $$
> > > \Delta = \text{Linear}_1(x)
> > > $$
> > > However, in [1], it is mentioned in the appendix section C that:
> > >  $$
> > > \Delta = \text{softplus(Linear}(x))
> > > $$
> > > Therefore,  $\Delta$ being a linear layer will be wrong assumption if the S6/Mamba model is being referenced here. It will be correct, though, if S4 model is being talked about here.
> > >
> > > 5. On line 165, the notation is changed from $x^{t-1}$ introduced in line 090 to $X^{t-1}$ .
> > > 6. Line 167 mentions "where $g^t$ is used to summarize the first $t$ layers" What is meant by "summarize" here? In the same line $A^t$ is mentioned but not mentioned what operator does it represent or if the suitable operator for this will be introduced later or not...
> > >
> > > 7. Important question from my side: On line 184 it is mentioned that resnets can be treated as a layer aggregation. By extension, transformers should also be treated as layer aggregators because transformers also use the same residual connections as in ResNets. Therefore, it seems redundant to me to do layer aggregation on layer aggregation.  In "Add & Norm layer", the Add part is the aggregating operator. Moreover, on line 209, how does MLP fit into the equation?
> > >
> > > 8. Lines 213 - 215 introduce a mathematical notation and it is bewildering for me to read suddenly about "In financial statistics, time series models with discrete states are employed for low-frequency data, while the diffusion model with continuous processes is a standard tool for high-frequency data." out of nowhere. This motivation feels forced on the reader rather than being a product of a proper investigation or underlying problem. Moreover, here "discrete states" actually make sense but "continuous processes" do not. Also, does "diffusion model" in this line mean a diffusion neural network or a model for a physical diffusion process?
> > >
> > >  9. $X^{t-1}$ in equation 12 should be $X^t$ as described by state-space equations, right? Also in line 266, 269 and 290. Following the same equation (12), there is no explanation of what $f^t$ is. It is vaguely touched upon in line 249.
> > >
> > > 10. In line 257, what is mentioned by state in the context of ResNets? And it is mentioned that $N$ is the dimension of latent states but which ones and how many?
> > >
> > > 11. There is no mathematical mention of "concatenation along feature dimension" except for maybe a partial mention of state in line 258 which itself is not clear if it refers to ResNets or SSM.
> > >
> > > 12. On line 292, $O^{t-1}$ is not defined.
> > >
> > > 13. On line 294, Why do you choose to define it this way? Even if it's obvious that it is inspired from resnets, or densenets, please mention it so that it doesn't seem like this is an arbitrary choice.
> > >
> > > 14. It's not clear why simply $\overline{A}$ and $\overline{B}$ isn't used from equation 4 and 5. In line 299, $W_{\Delta}$ is not defined. Similarly $W$ in equation 18 is not defined.
> > >
> > > 15. On line 316, it is mentioned that the tokens are split into two components, patch tokens and class token but there is no mention of what is the reason behind this.
> > >
> > > 16. There is no explanation or detailed description of S6LA module in any place other the caption of Figure 3.
> > >
> > >
> > >
> > >
> > >
> > >
> > >
> > >
> > >
> > >
> > >
> > > Very minor: line 178 "at Equation. 6" instead of "on" or maybe "in"?.
> > > Very very minor: hyphen missing in "$t$th layer" on line 215.
> > >
> > >
> > >
> > > Reference:
> > > 1. Gu, Albert, and Tri Dao. "Mamba: Linear-time sequence modeling with selective state spaces." arXiv preprint arXiv:2312.00752 (2023).

---

> > > > ### Author Response · Authors · 2024-11-25
> > > >
> > > > Thank you for your thorough review. We have carefully considered your comments and responded to each one individually. We greatly appreciate the time you took to review our article.
> > > >
> > > > **Questions 1 \& 2:**
> > > >
> > > > Thank you for addressing the continuous process problem and its motivation. We will present our answers from three perspectives:
> > > >
> > > > * **Continuous Process**: For example, if we consider a fixed time interval, such as one day, and sample stock prices every minute, we generate a large number of discrete data points that can be treated as continuous. Conversely, if we sample stock prices every hour, those points should be regarded as discrete. This illustrates that with sufficient sampling at regular intervals, stock prices sampled high frequently can be viewed as a continuous process.
> > > >
> > > > * **Deep neural networks**: From input to output, a deep neural network can also be viewed as an interval. Within this framework, having many layers in the network resembles how stock data behaves as a continuous process. For instance, ResNet-152, which contains 152 layers, can be represented as a continuous process (a smooth curve). This perspective is not entirely new; it has been discussed in previous works [1,2], although those works did not focus on layer interactions.
> > > >
> > > > * **Layer Interaction**: Previous research has produced numerous papers addressing layer interactions in deep neural networks [3,4]. However, these studies primarily focus on a discrete perspective. In contrast, this paper represents the first attempt to explore layer interactions through a continuous lens. By adopting this continuous perspective, we aim to uncover deeper insights into the relationships between layers.
> > > >
> > > > **Questions 3:**
> > > >
> > > > $x$ represents the output feature of one specific layer and $t$ is the index of one specific layer. Therefore, $x^t$ represents the output feature of the $t$-th layer.
> > > >
> > > > **Questions 4:**
> > > >
> > > > For the first one $\Delta=Linear(x)$ on Line 161, we keep the formulation consistent with the ones in S4 since we are revisiting the classic SSM. We appreciate your careful review on our manuscript, and we acknowledge that in S6/Mamba, it is reformulated as $\Delta=softplus(Linear(x))$, and we would like to clarify that we utilized this S6/Mamba formulation in our implementation of S6LA. We will make this point clear in our final version.
> > > >
> > > > **Questions 5:**
> > > >
> > > > Thanks for your notification, we have align the notations in Line 165 and Line 090 as $X^{t-1}$.
> > > >
> > > > **Questions 6:**
> > > >
> > > > The meaning of 'summerize' in Line 167, is that we will need to aggregate the output features from layer $0$ to layer $t$ (which are $X^0$ to $X^t$) as a unified feature $A^t$. In our paper, we used $g^t$ to denote this aggregation function/operation at layer $t$, and it could have different formulations. For instance, as detailed in Equation (7) Line 175, for DenseNet, the $g^t$ can be formulated as the concatenation of $X^0$ to $X^t$, which could be represented as $\text{Conv1}^t(\text{Concat}(\boldsymbol{X}^0, \boldsymbol{X}^1, \ldots, \boldsymbol{X}^{t-1}))$.
> > > >
> > > > **Questions 7:**
> > > >
> > > > Thank you for your question about layer aggregation in transformer. We acknowledge that the Transformers' built-in residual connection could also be considered as a **naive** layer aggregation (as we detailed in Section 3.3). However, we assume such simple residual connection is not effective enough to aggregate the complement information from all previous ViT blocks. Therefore, in our study, we proposed S6LA to store the complete information within the hidden state, and interact between ViT blocks using a multiplication operation that similar to the attention machanism. In addition, experimental results also suggest that S6LA is **not redundant**. For instance, S6LA leads to nearly a $1.5\%$ improvement in Top-1 accuracy on ImageNet classification as shown in Table 2 in our manuscript. For the MLP in Line 209, it is in the function of $g^t$.
> > > >
> > > > **Questions 8:**
> > > >
> > > > For the confusion about the "continuous process in finacial statistics", we have provied a more detailed explaination in response to Question 1. In addition, the definition of diffusion model mentioned in Line 213-215 is not the diffusion neural network or physical diffusion process. It is a concept in statistics. To prevent the ambiguity, we have revised this section and explained it in more details.
> > > >
> > > > **Questions 9:**
> > > >
> > > > 1. Thanks for your notification and careful review, we have corrected the $X^{t-1}$ in Equation (12) to $X^t$.
> > > > 2. We did not explain the notation of $f^t$ in Equation (12) since it is actually the same as the notation given in previous Section 3.2 in Line 167: $f^t$ produces the new layer output from the last hidden layer and the given aggregation which contains the previous information.

---

> > > > > ### Author Response · Authors · 2024-11-25
> > > > >
> > > > > **Questions 10:**
> > > > >
> > > > > In the context of ResNets, "state" typically refers to the intermediate representations of the input data as it passes through the layers of the network. It is composed of some combinations of 3\*3 Convolution Layers and 1\*1 Convolution Layers. $N$ is the dimension of latent state given in our S6LA model. And it is a hyperparameter which is discussed in ablation study in Table 6, which we tried 16, 32, and 64.
> > > > >
> > > > > **Questions 11:**
> > > > >
> > > > > The meaning of "concatenation along feature dimension" is we concate the dimension of features from the $X$ and $h$, which means the dimension of $X$ is $R ^ {H \times W \times D}$, $h$ is $R ^ {H \times W \times N}$, the dimension of concatenation is $R ^ {H \times W \times (D+N)}$.
> > > > >
> > > > > **Questions 12:**
> > > > >
> > > > > The definition of $O^t$ is the temporary variable which is the concatenation of $X$ and $h$.
> > > > >
> > > > > **Questions 13:**
> > > > >
> > > > > We would like to clarify that our idea is mainly based on SSM instead of ResNets/DenseNets, therefore, we need to keep a hidden state to store the information from previous features and interact with the features from the current layer, and also update the hidden state in each layer aligning with the sequence processing mechanism in SSM.
> > > > >
> > > > > **Questions 14:**
> > > > >
> > > > > For the parameter $\bar{A}$ and $\bar{B}$, since our assumption is from SSM [5], and the state space model is defined as $ h^{\prime}(t) =A h(t)+B x(t)$, where it maps a $1$-dimensional input signal $x(t)$ to an $N$-dimensional latent state $h(t)$.
> > > > >
> > > > > Then follow S4 [5] and S6 [6], using ZOH (zero-order hold) condition, the continuous process SSM can be discreted by $ h^{t} = e^{\Delta A}h^{t-1} + (\Delta A)^{-1}(\text{exp}(\Delta A)-I) \cdot \Delta B x^{t}$.
> > > > >
> > > > > In S4 [5] and S6 [6], Albert uses $\overline{A} = \text{exp}(\Delta A)$ and $\overline{B} = (\Delta A)^{-1}(\text{exp}(\Delta A)-I) \cdot \Delta B$ to simplify the above complex notations.
> > > > >
> > > > > With the dirst-order Taylor series, Albert also lets $\overline{B} \approx \Delta B$.
> > > > >
> > > > > Therefore, we cannot simply use these two parameters $\bar{A}$ and $\bar{B}$ because our initial conditions are based on the state space model (SSM). These two parameters are generated through discretization of the original parameters, so they cannot be used directly.
> > > > >
> > > > > If we simply use $ \bar{A} $ and $ \bar{B} $, it would be equivalent to an RNN, which does not stem from the perspective of the state space model (SSM). Therefore, we cannot rely on these simple parameters.
> > > > >
> > > > > In deep learning network, $W_{\Delta}$ is the linear projection of $\Delta$ and similar, $W$ is the linear projection.

---

> > > > > > ### Author Response · Authors · 2024-11-25
> > > > > >
> > > > > > **Questions 15:**
> > > > > >
> > > > > > In the standard Vision Transformer (ViT) architecture [7], patch tokens and the class token are crucial components that facilitate the processing of image data. Here's an overview description of each and more details could be found in the original ViT paper [7]:
> > > > > >
> > > > > > Patch Tokens:
> > > > > > 1. Definition: In ViT, an image is divided into fixed-size non-overlapping patches. Each patch is then flattened into a vector, which is referred to as a patch token.
> > > > > > 2. Processing: These patch tokens serve as the input tokens to the transformer model. For example, if an image is divided into 16\*16 pixel patches, each patch is flattened into a vector and embedded into a lower-dimensional space using a linear projection.
> > > > > > 3. Purpose: This approach allows the model to treat the image as a sequence of tokens (similar to words in NLP), enabling the application of transformer architectures designed for sequential data.
> > > > > >
> > > > > > Class Token:
> > > > > > 1. Definition: The class token is an additional learnable token that is prepended to the sequence of patch tokens. It acts as a representative for the entire image.
> > > > > > 2. Purpose: After the transformer processes the sequence of patch tokens (including the class token), the output corresponding to the class token is used for classification tasks. This allows the model to aggregate information from all patches to make predictions about the entire image.
> > > > > >
> > > > > > **Questions 16:**
> > > > > >
> > > > > > For the details of S6LA, we only keep the detailed caption for Figure 3 due to the page limitation. The full operation is described using the equations in Section 4.2 and 4.3, Specifically:
> > > > > >
> > > > > > 1. For the details of **Deep CNNs**:
> > > > > >
> > > > > >     1. **Input Treatment**: we concate the representation $\boldsymbol{X}^{t-1}$ and $h^{t-1}$:
> > > > > >     $$
> > > > > >     \boldsymbol{O}^{t-1} = Concat(\boldsymbol{X}^{t-1},h^{t-1}).
> > > > > >     $$
> > > > > >     2. **Update of Latent State**:
> > > > > >     $$
> > > > > >     h^t = e^{(\Delta A)} h^{t-1} + \Delta B \boldsymbol{O}^{t-1}.
> > > > > >     $$
> > > > > >     3. **Update of Input State**:
> > > > > >     $$
> > > > > >     \boldsymbol{X}^{t} = \boldsymbol{O}^{t-1} + \boldsymbol{X}^{t-1}.
> > > > > >     $$
> > > > > >
> > > > > > 2. For the details of **Deep ViTs**:
> > > > > >
> > > > > >     1. **Input Treatment**: we begin by combining the class token and input, alongside the application of positional embeddings.
> > > > > >     $$
> > > > > >     \boldsymbol{X}^{t-1}_{\text{input}} = \text{Add} \& \text{Norm}(\text{MLP}(\text{Add} \& \text{Norm}(\text{Attn}(\boldsymbol{X}^{t-1})))),  \quad
> > > > > >     \boldsymbol{X}^{t-1}_p, \boldsymbol{X}^{t-1}_c = \text{Split}(\boldsymbol{X}^{t-1}_{\text{input}}).
> > > > > >     $$
> > > > > >
> > > > > >     2. **Latent State Update**:
> > > > > >     $$
> > > > > >     h^t = e^{(\Delta A)} h^{t-1} + \Delta B \boldsymbol{X}^{t-1}_c
> > > > > >     $$
> > > > > >
> > > > > >     3. **Output Computation**:
> > > > > >     $$
> > > > > >     \widehat{\boldsymbol{X}}_p^{t-1} = \boldsymbol{X}^{t-1}_p + W\boldsymbol{X}^{t-1}_p h^t, \\
> > > > > >     \boldsymbol{X}^t = \text{Concat}(\widehat{\boldsymbol{X}}_p^{t-1},\boldsymbol{X}^{t-1}_c).
> > > > > >     $$
> > > > > > We appreciate your suggestion on the detailed caption of specifica S6LA, and we have updated Figure X and Figure X in Appendix to improve our clarity.
> > > > > >
> > > > > > We have incorporated the revisions mentioned above into our manuscript and highlighted with **blue**. We hope that our updated responses effectively address the concerns raised by you. We truly appreciate the opportunity to improve our manuscript based on your valuable feedback and look forward to any further discussions you may wish to have with us.
> > > > > >
> > > > > >
> > > > > > Reference:
> > > > > >
> > > > > > 1. Alejandro F Queiruga, N Benjamin Erichson, Dane Taylor, and Michael W Mahoney. Continuous-in-depth neural networks. arXiv preprint arXiv:2008.02389, 2020.
> > > > > >
> > > > > > 2. Xuanqing Liu, Tesi Xiao, Si Si, Qin Cao, Sanjiv Kumar, and Cho-Jui Hsieh. How does noise help
> > > > > > robustness? explanation and exploration under the neural sde framework. In Proceedings of the
> > > > > > IEEE/CVF conference on computer vision and pattern recognition, pp. 282–290, 2020.
> > > > > >
> > > > > > 3. Yanwen Fang, Yuxi Cai, Jintai Chen, Jingyu Zhao, Guangjian Tian, and Guodong Li. Cross-layer
> > > > > > retrospective retrieving via layer attention. arXiv preprint arXiv:2302.03985, 2023.
> > > > > >
> > > > > > 4. Jingyu Zhao, Yanwen Fang, and Guodong Li. Recurrence along depth: Deep convolutional neural
> > > > > > networks with recurrent layer aggregation. Advances in Neural Information Processing Systems,
> > > > > > 34:10627–10640, 2021.
> > > > > >
> > > > > > 5. Albert Gu, Karan Goel, and Christopher R´e. Efficiently modeling long sequences with structured state spaces. arXiv preprint arXiv:2111.00396, 2021a.
> > > > > >
> > > > > > 6. Albert Gu and Tri Dao. Mamba: Linear-time sequence modeling with selective state spaces. arXiv
> > > > > > preprint arXiv:2312.00752, 2023.
> > > > > >
> > > > > > 7. Alexey Dosovitskiy. An image is worth 16x16 words: Transformers for image recognition at scale. arXiv preprint arXiv:2010.11929, 2020.

---

> > > > > > > ### Comment · Reviewer_nBY3 · 2024-11-25
> > > > > > >
> > > > > > > Thank you for your patience and for taking the time to answer each of my concerns carefully. It has cleared up some misconceptions that arose due to the writing of the paper. The purpose of putting forward my questions was to be three-pronged: To help me understand better if I have misunderstood something, or if there is some mistake then it is corrected and improving on text considering the reader's perspective. The mistakes that I have pointed out or the clarifications that I asked for are meant to make the manuscript easier to understand.  I am hopeful that the revised manuscript will be easier for the readers with all the corrections made.
> > > > > > >
> > > > > > > Regarding questions 1&2: if I understand correctly, a large number of data points can be treated as continuous. But what exactly has changed (mathematically or otherwise) by treating the process as continuous in your work?
> > > > > > >
> > > > > > >
> > > > > > > In light of your responses and having understood the work better, I am increasing my rating.

---

> > > > > > > > ### Author Response · Authors · 2024-11-25
> > > > > > > >
> > > > > > > > Thank you for your patience about taking the time to discuss with us about our paper. It is really helpful for us.
> > > > > > > >
> > > > > > > > We really appreciate all your comments and suggestions, which lead to a substainly improved version, and the presentation of this paper has also become clearer and clearer.
> > > > > > > >
> > > > > > > > For the question:
> > > > > > > >
> > > > > > > > Generally speaking, the difference exactly is that between RNN and SSM. Specifically, if we treat the deep neural network as discrete, then the $\overline{A}$ and $\overline{B}$ at equation 4 in Line 155 will be the **trainable weights**, as you mentioned in Question 14 in your previous comments. This actually corrspond to RNN with linear activation (which is utilized in RLA [1]). On the other hand, if we treat the deep neural network as continuous, then $\overline{A}$ and $\overline{B}$ will have the forms in Equations 4 and 5 at lines 155 and 158 respectively.
> > > > > > > >
> > > > > > > > More specifically, for the **exact mathematical changes**, the RLA [1] method with the discrete perspective of layer interaction has been formulated as:
> > > > > > > >
> > > > > > > > $$
> > > > > > > > h^t = g^t(h^{t-1},\boldsymbol{X}^{t}) \quad and \quad \boldsymbol{X}^t = f^t(h^{t-1},\boldsymbol{X}^{t}),
> > > > > > > > $$
> > > > > > > > where $g^t$ is the different forms which achieve various RLA modules to help the main CNN better extract features and weight sharing can be further used to regularize the layer aggregation. The form of $f^t$ is mainly determined by the currently used mainstream CNNs.
> > > > > > > > The update can be formulated as:
> > > > > > > >
> > > > > > > > $$
> > > > > > > > h^t = W Concat(\boldsymbol{X}^{t}, h^{t-1}) + V h^ {t-1} \quad and \quad \boldsymbol{X}^{t+1} = Concat(\boldsymbol{X}^{t}, h^{t-1}) + \boldsymbol{X}^{t}
> > > > > > > > $$
> > > > > > > > where $W$ is the linear projection of Convolutional layer $W = Linear_{W}(Conv())$.
> > > > > > > >
> > > > > > > > In contrast, our proposed S6LA has a similar yet fundamentally different formulation:
> > > > > > > >
> > > > > > > > 1. **Input Treatment**: we concate the representation $\boldsymbol{X}^{t}$ and $h^{t-1}$:
> > > > > > > > $$
> > > > > > > > \boldsymbol{O}^{t} = Concat(\boldsymbol{X}^{t},h^{t-1}).
> > > > > > > > $$
> > > > > > > > 2. **Update of Latent State**:
> > > > > > > > $$
> > > > > > > > h^t = \overline{A} h^{t-1} + \overline{B} \boldsymbol{O}^{t}.
> > > > > > > > $$
> > > > > > > > where $\overline{A} = \text{exp}(\Delta A)$ and $\overline{B} = \Delta B$,
> > > > > > > >
> > > > > > > > and $\Delta = W_{\Delta} (\text{Conv}(\boldsymbol{O}^{t})), B = W_B (\text{Conv}(\boldsymbol{O}^{t}))$ which means the selective mechanism.
> > > > > > > >
> > > > > > > > 3. **Update of Input State**:
> > > > > > > > $$
> > > > > > > > \boldsymbol{X}^{t+1} = \boldsymbol{O}^{t} + \boldsymbol{X}^{t}.
> > > > > > > > $$
> > > > > > > >
> > > > > > > > Therefore, the exact mathematical modifications of our method compared to the discrete perspective method (RLA) are as follows: $ \overline{A} $ and $ \overline{B} $ take the forms specified in Equations 4 and 5 through discretization, rather than being simple trainable weights. They depend on $\Delta$, $ A $, and $ B $.
> > > > > > > >
> > > > > > > > The effectiveness of this practice can be found by experimental results which are shown as followings:
> > > > > > > >
> > > > > > > > 1. Classification task on ImageNet:
> > > > > > > >
> > > > > > > > | Model | Top-1 ACC. | Top-5 ACC. |
> > > > > > > > | :---: | :---: | :---: |
> > > > > > > > |ResNet-50 | 76.1 | 92.9 |
> > > > > > > > |+ RLA (discrete) | 77.2 | 93.4 |
> > > > > > > > |+ S6LA (continuous) | 78.0| 94.2 |
> > > > > > > >
> > > > > > > > 2. Object detection task on MS COCO 2017 (Detector: Faster R-CNN):
> > > > > > > >
> > > > > > > > | Model | Detector | $AP^{bb}$ | $AP^{bb}_{50}$ | $AP^{bb}_{75}$ | $AP^{bb}_{S}$ | $AP^{bb}_{M}$ | $AP^{bb}_{L}$ |
> > > > > > > > | :---: | :---: | :---: | :---: | :---: | :---: | :---: | :---: |
> > > > > > > > |ResNet-50 | Faster R-CNN | 36.4 | 58.2 | 39.2 | 21.8 | 40.0 | 46.2 |
> > > > > > > > |+ RLA (discrete) | Faster R-CNN | 38.8 | 59.6 | 42.0 | 22.5 | 42.9 | 49.5 |
> > > > > > > > |+ S6LA (continuous) | Faster R-CNN | 40.3 | 61.7 | 43.8 | 24.2 | 44.0 | 52.5|
> > > > > > > >
> > > > > > > > 3. Object detection task on MS COCO 2017 (Detector: RetinaNet):
> > > > > > > >
> > > > > > > > | Model | Detector | $AP^{bb}$ | $AP^{bb}_{50}$ | $AP^{bb}_{75}$ | $AP^{bb}_{S}$ | $AP^{bb}_{M}$ | $AP^{bb}_{L}$ |
> > > > > > > > | :---: | :---: | :---: | :---: | :---: | :---: | :---: | :---: |
> > > > > > > > |ResNet-50 | RetinaNet | 35.6 | 55.5 | 38.2 | 20.0 | 39.6 | 46.8 |
> > > > > > > > |+ RLA (discrete) | RetinaNet | 37.9 | 57.0 | 40.8 | 22.0 | 41.7 | 49.2 |
> > > > > > > > |+ S6LA (continuous) | RetinaNet | 39.3 | 59.0 | 41.9 | 23.7 | 42.9 | 51.0|
> > > > > > > >
> > > > > > > > 4. Object detection and instance segmentation task on MS COCO 2017 with Mask R-CNN:
> > > > > > > >
> > > > > > > > | Model | $AP^{bb}$ | $AP^{bb}_{50}$ | $AP^{bb}_{75}$ | $AP^{m}$ | $AP^{m}_{50}$ | $AP^{m}_{75}$ |
> > > > > > > > | :---: | :---: | :---: | :---: | :---: | :---: | :---: |
> > > > > > > > |ResNet-50 | 37.2 | 58.9 | 40.3 | 34.1 | 55.5 | 36.2 |
> > > > > > > > |+ RLA (discrete) | 39.5 | 60.1 | 43.4 | 35.6 | 56.9 | 38.0 |
> > > > > > > > |+ S6LA (continuous) | 40.6 | 61.5 | 44.2 | 36.7 | 58.3 | 38.3|
> > > > > > > >
> > > > > > > > Therefore, based on the experimental results, we believe our approach using a continuous perspective outperforms the discrete perspective in previous works.
> > > > > > > >
> > > > > > > > Thank you for your patient discussion with us. If you have any further questions about this paper, we welcome the opportunity to discuss them with you.
> > > > > > > >
> > > > > > > > Reference:
> > > > > > > >
> > > > > > > > [1] Jingyu Zhao, Yanwen Fang, and Guodong Li. Recurrence along depth: Deep convolutional neural
> > > > > > > > networks with recurrent layer aggregation. Advances in Neural Information Processing Systems,
> > > > > > > > 34:10627–10640, 2021.

---

> > > > > > > > > ### Author Response · Authors · 2024-11-27
> > > > > > > > >
> > > > > > > > > Dear Reviewer nBY3,
> > > > > > > > >
> > > > > > > > > I hope this message finds you well. I am writing to follow up on the review of our manuscript. We greatly appreciate your valuable feedback and the time you have taken to consider our work.
> > > > > > > > >
> > > > > > > > > We understand that you may have a busy schedule, but we are eager to receive your further comments and suggestions at your earliest convenience. Your prompt response would be greatly appreciated and would enable us to make the necessary revisions in a timely manner. And we are eager to address any further comments or questions from you to improve our work.
> > > > > > > > >
> > > > > > > > > Thank you once again for your insightful review. We look forward to hearing from you soon.
> > > > > > > > >
> > > > > > > > > Best regards,
> > > > > > > > > Authors

---

> > > > > > > > > > ### Author Response · Authors · 2024-12-02
> > > > > > > > > > **The deadline of discussion**
> > > > > > > > > >
> > > > > > > > > > Dear Reviewer nBY3,
> > > > > > > > > >
> > > > > > > > > > I hope this message finds you well. With the rebuttal deadline approaching in the last two days, we would greatly appreciate any feedback or insights you might have.
> > > > > > > > > >
> > > > > > > > > > We understand that the review process can be demanding, and we genuinely value the time and effort you invest. Your input is crucial to us, and we are eager to address any comments or questions to enhance our work.
> > > > > > > > > >
> > > > > > > > > > Thank you once again for your time and consideration. We look forward to your response soon.
> > > > > > > > > >
> > > > > > > > > > Best regards,
> > > > > > > > > > Authors of Submission 2351

---

### Meta-Review · Area_Chair_zYDF · 2024-12-20

**Metareview:**

This paper focuses on improving the representational power of advanced vision networks by proposing a novel layer aggregation strategy called Selective State Space Model Layer Aggregation (S6LA). The method integrates traditional CNN and Transformer architectures into a sequential framework. The paper evaluates the approach on image classification, object detection, and segmentation benchmarks.

**Additional Comments On Reviewer Discussion:**

The authors did a good job elaborating on the concepts and motivation behind formulating the neural network through the lens of continuous processing. However, some minor concerns remain regarding the somewhat heuristic choice between concatenation and multiplication in the CNN and Transformer-based methods. That said, the authors have conducted comprehensive experiments to evaluate these designs, which generally support their statements. Incorporating all the new results and clarifications provided during the rebuttal phase will further enhance the completeness of the paper.

---

### Decision · Program_Chairs · 2025-01-22

Accept (Poster)